

# Study of the threshold for the POT method based on hindcasted significant wave heights of tropical cyclone waves in the South China Sea

Zhuxiao Shao[1], Bingchen Liang[1, 2], Huajun Li[1, 2], Ping Li[3], Dongyoung Lee[1, 4]

[1] College of Engineering, Ocean University of China, 238 Songling Road, Qingdao 266100, China

[2] Shandong Province Key Laboratory of Ocean Engineering, Ocean University of China, 238 Songling Road, Qingdao 266100, China

[3] China Classification Society, Beijing 100007, China

[4] Korea Institute of Ocean Science & Technology, Ansan, Korea

*Corresponding to*: Bingchen Liang (bingchen@ouc.edu.cn)

**ABSTRACT**

An assessment of extreme significant wave heights is performed in the South China Sea (SCS), which is crucial for the coastal and offshore engineering in this area. Two significant factors influencing the assessment are the initial database and the assessing method. The initial database is a basic for assessment, and the assessing method is used to extrapolate appropriate return significant wave heights based on this database during a period. In this

study, a 40-year (1975-2014) hindcasted significant wave height of tropical cyclone waves is adopted as the initial database. Based on this database, the peak significant wave height of every tropical cyclone wave is directly extracted as the initial sample; the independent and identically distributed assumption is satisfied; and the interference for the selection of the sample is avoided. The peak over threshold (POT) method with the generalized Pareto distribution (GPD) model is employed to extract the sufficiently large and high sample for

model estimation. The peak excesses over a sufficiently high value (i.e., threshold) are fitted; thus, the return

significant wave heights are highly dependent on the threshold. To determine the unique threshold for the POT method, characteristics of tropical cyclone waves are researched. The research results reveal that the separation value shown in the distribution of the initial sample is suitable for sampling in the SCS. Because the separation value is within the stable threshold range and the asymptotic tail approximation and estimation uncertainty are

reasonable, the selected threshold is suitable and the corresponding return significant wave height is reliable.

## 1. Introduction

Reasonable assessment of extreme significant wave heights is highly important for the security and cost of coastal defences and offshore structures (Ojeda and Guillén, 2006, 2008; Ojeda et al., 2010, 2011; Mortlock and Goodwin, 2015, 2016; Mortlock et al., 2017). To obtain this assessment, an appropriate probability distribution

model is fitted based on a stable sample, which is extracted from an accurate initial database by a reliable sampling method.

The initial database is a basic for assessment of extreme significant wave heights (Godoi et al., 2017; Lucas et al., 2017; Li et al., 2018). In previous studies, the long-term continuous database is usually employed as the initial database, such as a 32-year measured significant wave height in the Gulf of Maine (Viselli et al., 2015), a 44-year

hindcasted significant wave height in the North Atlantic Ocean (Muraleedharan et al., 2016) and a 22-year hindcasted significant wave height in the Yellow Sea (Gao et al., 2018). Considering that the extreme significant wave height is extrapolated based on an independent and identically distributed sample required for the extreme value theory (EVT) (Coles, 2001; Sobradelo et al., 2011), these time series buoy measurements and numerical hindcasts are processed before sampling. The homogenous methodology is used to extract homogenous

significant wave heights via separation in carefully chosen directional sectors and seasonal analyses as well as



separation of the sea state into independent wave systems (Lerma et al., 2015; Solari and Alonso, 2017). The declustering methodology, such as the double-threshold approach (Mazas and Hamm, 2011) and the minimum separation time method (Kapelonis et al., 2015), is used to differentiate the individual wave event. After implementing these methodologies, the initial sample is independently extracted under the same type of

meteorological event, and the sample can be further extracted from the initial sample. However, these methodologies introduce uncertainty in the initial sample (such as the subjectivity of practitioners in the selections of the initial threshold and time window), which influences the selection of the sample.

Typical approaches used for assessment of extreme significant wave heights are that the probability distribution model constructs long-term distributions based on samples extracted by the sampling method (Muraleedharan et

al., 2016). To obtain the sufficiently large and high sample for model estimation, the peak over threshold (POT) method (Goda et al., 2001) is used to identify the peak significant wave heights over a threshold (Ferreira and Guedes Soares, 1998; Soares and Scotto, 2004; Caires and Sterl, 2005; Benetazzo et al., 2012; You and Callaghan, 2013; Xiao et al., 2017). Additionally, the generalized Pareto distribution (GPD) model (Coles, 2001) is used to extrapolate extreme significant wave heights based on the sample extracted by the POT method

(Martucci et al., 2010; Blanchet et al., 2015; Kapelonis et al., 2015; Boessenkool et al., 2017; Muhammed Naseef and Sanil Kumar, 2017). This method (i.e., POT/GPD method) (Alves and Young, 2003; You, 2011; Vanem, 2015a; Samayam et al., 2017; Shao et al., 2017) takes the peak significant wave heights above a threshold as the sample to adjust the parametric distribution. Therefore, the predicted results are highly dependent on the threshold. In previous studies, many threshold selection methods have been proposed, such as graphical

diagnostics (Coles, 2001; Sánchez-Arcilla et al., 2008; Bernardara et al., 2014), empirical methods (Ferreira et al., 2003; Neves and Alves, 2004; Reiss and Thomas, 2007), probabilistic-based techniques (Hill, 1975; Beirlant et al., 2006; Goegebeur et al., 2008), computational approaches (Danielsson et al., 2001; Beguería, 2005; Solari et

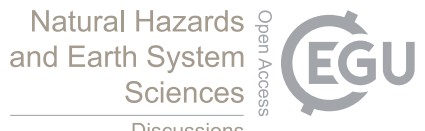

al., 2017) and mixture models (Carreau and Bengio, 2009; Eastoe and Tawn, 2010; MacDonald et al., 2011). Among these methods, a graphical diagnostic referred to as the sensitivity of the return significant wave height to the threshold (Scarrott and MacDonald, 2012) is commonly accepted (Petrov et al., 2013; Northrop and Coleman, 2014; Vanem, 2015b; Northrop et al., 2017; Sulis et al., 2017). This method fits the GPD over a range of

candidate thresholds and selects the suitable threshold through identifying stability of return significant wave heights. If return significant wave heights are insensitive to the threshold, the corresponding threshold can be selected as the suitable threshold. The benefit of this method is that it requires practitioners to graphically inspect and comprehend the features of data and assess the uncertainty of the candidate thresholds (Scarrott and MacDonald, 2012). The drawback of this method is that the threshold is not uniquely selected, and another

criterion is needed to identify the optimal one (Lerma et al., 2015).

In the South China Sea (SCS), time series wave parameters have been simulated (Zheng et al., 2012; Mirzaei et al., 2015; Yaakob et al., 2016), and extreme waves have been investigated based on long-term continuous data (Zheng et al., 2015; Chen et al., 2017; Wang et al., 2018). The annual maxima (AM) method (Tawn, 1988) is usually employed to extract the annual maximal significant wave height as the sample for extrapolation.

Considering that the sampling method is a basic for assessment of extreme significant wave heights, Shao et al., (2018a) compare the AM method with the POT method. They find that the distribution and representative of samples in the SCS limit the application of the AM method. Even though the return period is close to the size of the dataset, the sample of the AM method may be unreasonable for extrapolation. Because the POT is a natural sampling method without addition limitation and it guarantees the number and representative of samples when

the threshold is suitable, Shao et al., (2018a) suggest that the POT method is more suitable for sampling in the SCS. In this study, the POT method is further studied in the SCS based on this conclusion. Before assessing the extreme significant wave height in the SCS, the meteorological analysis is implemented. The tropical cyclone is a





major extreme weather in the SCS, which always drives the storm wave (Anoop et al., 2015; Hithin et al., 2015; Sanil Kumar and Anoop, 2015; Ojeda et al., 2017; Wang et al., 2017; Mortlock et al., 2018; Sanil Kumar et al., 2018). In addition, the number of tropical cyclones is counted in the SCS, which shows that the tropical cyclone wave is feasible for studying extreme significant wave heights. Thus, extreme significant wave heights are

extrapolated based on tropical cyclone waves, and 40-year (1975-2014) hindcasted significant wave heights obtained during tropical cyclones are employed as the initial database. Because this initial database is independently simulated during the tropical cyclone, the peak significant wave height of every tropical cyclone wave is directly extracted as the initial sample, and this initial sample satisfies the independent and identically distributed assumption. Moreover, the process of the assessment is simplified, and the sample can be extracted

without the influence of the homogenous and declustering methodologies. After determining the initial sample, the POT method is used to extract the peak significant wave heights over the threshold as the sample. Shao et al., (2018a) and Liang et al., (2019) have analysed the sensitivity of the return significant wave height to the threshold for threshold selection. They found that the suitable threshold should be determined within the stable threshold range (i.e., a threshold range corresponding to a range of stable return significant wave heights).

However, a unique threshold cannot be identified by this method. To select the unique threshold within the stable threshold range, Shao et al., (2018a) defined the highest threshold within the common stable threshold range as the suitable threshold. They preliminarily studied threshold selection of the POT method to analyse the sampling methods (i.e., the POT method and AM method) in the SCS. Liang et al., (2019) focused on threshold selection of the POT method and proposed an automated threshold selection method based on the characteristic of

extrapolated significant wave heights (ATSME). The ATSME employs the differences in extrapolated significant wave heights for neighbouring thresholds as the diagnostic parameters to identify the uniquely stable threshold range via an automated technique and selects the highest threshold within the stable threshold range as the

suitable threshold for different return periods. This method can select a suitable threshold by a pragmatically automated and computationally inexpensive technique. Considering that threshold selection criteria of Shao et al., (2018a) and Liang et al., (2019) are based on the sensitivity of the return significant wave height to the threshold, a suitable threshold cannot directly be selected without a subjective definition, which may be different for

different practitioners. Liang et al., (2019) diagnosed the return significant wave height within the stable threshold range. If some return significant wave heights within the stable threshold range are relatively different from the others, the corresponding candidate thresholds are rejected. Thus, the influence of the subjective definition for the threshold and return significant wave height is weakened. This diagnostic process is crucial for the assessment of extreme wave heights, especially for the regional assessment of extreme significant wave

heights. However, the subjective definition still exists in the ATSME. In the present study, threshold selection of the POT method is further studied based on the characteristic of the tropical cyclone wave in the SCS. The study results reveal that the separation value shown in the distribution of the initial sample is suitable for sampling. To validate the thresholds obtained by the distribution of the initial sample, the asymptotic tail approximation and estimation uncertainty are analysed, which show that the capabilities of this method for threshold selection. After

determining the sample, the GPD model is used to extrapolate extreme significant wave heights in the SCS.

The article is structured as follows. In the next section, the EVT including the sampling method and probability distribution model is introduced. Initial data and study sites are described in Section 3. In Section 4, initial samples are extracted, and the sensitivity of the return significant wave height to the threshold is discussed. Section 5 studies characteristics of tropical cyclone waves and the distribution of the initial sample. Finally, the

conclusions are presented in Section 6.



## 2. Background

The POT method extracts the peak significant wave heights above a selected value (i.e., threshold), $u$, as the sample.

For a threshold, $u$, that is sufficiently high, the distribution function of peak excesses over the threshold can be

5     approximated by a member of the GPD (Pickands, 1975; Embrechts et al., 1997):

$$F_u\left(Hs^*\right) = \begin{cases} 1-(1+k\dfrac{Hs^*}{\sigma})^{-\frac{1}{k}} & k \neq 0 \\[2mm] 1-\exp\left(-\dfrac{Hs^*}{\sigma}\right) & k = 0 \end{cases} \tag{1}$$

where $Hs^*$ represents peak excess over the threshold; $\sigma$ represents the scale parameter; and $k$ represents the shape parameter. These GPD parameters ($\sigma$ and $k$) are estimated using the maximum likelihood estimation method, which is recommended by Mazas and Hamm (2011):

10     $$\ln L(k,\sigma;Hs) = \begin{cases} -N\ln\sigma + (\dfrac{1}{k}-1)\displaystyle\sum_{j=1}^{N}\ln(1-\dfrac{kHs_j}{\sigma}) & k \neq 0 \\[3mm] -N\ln\sigma - \dfrac{1}{\sigma}\displaystyle\sum_{j=1}^{N}Hs_j, & k = 0 \end{cases} \tag{2}$$

where $N$ represents the number of events exceeding the threshold (i.e., the number of samples), and $Hs$ represents the peak significant wave height.

The return significant wave height for the $i$-year, $Hs_i$, is defined as follows:



$$Hs_i = F_u^{-1}(1 - \frac{1}{i})$$ (3)

Thus, it can be calculated with the following equation:

$$Hs_i = \begin{cases} u + [(\frac{N}{N_T}i)^k - 1]\sigma / k & k \neq 0 \\ u + \sigma \ln(\frac{N}{N_T}i) & k = 0 \end{cases}$$ (4)

where $N_T$ represents the size of the dataset.

**3. Initial data and study sites**

**3.1 Initial data**

As required by the EVT, the extreme significant wave height is extrapolated based on the independent wave

under the same type of meteorological event (Lerma et al., 2015; Solari and Alonso, 2017). Before assessing the

extreme significant wave height, the meteorological analysis is needed to identify the extreme weather. In the

10 SCS, the tropical cyclone frequently occurs, and the relatively high wave usually appears during the tropical

cyclone (Shao et al., 2017). It means that the tropical cyclone wave represents the extreme wave in the SCS well

and the extreme significant wave height can be assessed based on the tropical cyclone wave. Therefore,

significant wave heights from a 40-year (1975-2014) hindcast of tropical cyclone waves (Shao et al., 2018a) are

adopted as the initial database, which is simulated using the third-generation spectral wind-wave model SWAN

(an acronym for Simulating WAves Nearshore) (Booij et al., 1999; Mortlock et al., 2014; Amrutha et al., 2016).

This model is forced by the blended wind, which is obtained by combining the European Centre for



Medium-Range Weather Forecasts reanalysis wind and the Holland model wind (Shao et al., 2018b). Nine hundred and seventy-four tropical cyclone waves are independently simulated during the tropical cyclone. The spatial resolution is 0.0625 ° for both longitude and latitude, and the temporal resolution is 1 h.

## 3.2 Study sites

In this study, 22 locations are selected as the study site. Detailed information on latitude and longitude for the study site and the number of tropical cyclones recorded at the study site is shown in Table 1. A tropical cyclone is recorded at the study site when the distance between the centre of this tropical cyclone and the study site is within 300 km. Hourly significant wave heights simulated during the recorded tropical cyclones are adopted as the initial database at the study site.

To study the feasibility of the tropical cyclone wave for extrapolating extreme significant wave heights, the number of recorded tropical cyclones (this number determines the number of initial samples) is analysed at the 22 study locations. The number of recorded tropical cyclones is 247 to 403, and the annual mean number of recorded tropical cyclones is 6.175 to 10.075. The corresponding tropical cyclone waves are sufficient for assessment of extreme significant wave heights (Mazas and Hamm, 2011). To present this assessment and study characteristics

of tropical cyclone waves in detail, location #1 (22.00 °N, 118.75 °E) is selected as a representative.



Table 1
Study locations and numbers of tropical cyclones.

| Location | Latitude (°N) | Longitude (°E) | Number of Tropical Cyclones |
|---|---|---|---|
| #1 | 22 | 118.75 | 328 |
| #2 | 22.75 | 116.5 | 291 |
| #3 | 19.5 | 119 | 376 |
| #4 | 20.5 | 117.25 | 372 |
| #5 | 21 | 115.5 | 347 |
| #6 | 21.75 | 114.75 | 321 |
| #7 | 16 | 118 | 395 |
| #8 | 17 | 117 | 394 |
| #9 | 18 | 116.5 | 390 |
| #10 | 19 | 115.5 | 328 |
| #11 | 20 | 114 | 361 |
| #12 | 21.25 | 112.5 | 314 |
| #13 | 16 | 116 | 402 |
| #14 | 17.5 | 114.75 | 403 |
| #15 | 19 | 113.5 | 377 |
| #16 | 20 | 112 | 314 |
| #17 | 16.25 | 112.25 | 341 |
| #18 | 18 | 111 | 334 |
| #19 | 20 | 108.5 | 257 |
| #20 | 16.25 | 110 | 292 |
| #21 | 17.5 | 108.75 | 275 |
| #22 | 18.5 | 107.5 | 247 |

## 4. Study of the POT method

5  **4.1 Initial samples**

The initial sample needs to satisfy the independent and identically distributed assumption. Considering that the

initial database is only simulated during the tropical cyclone and comes from different tropical cyclone waves,

the peak significant wave height of the tropical cyclone waves is directly extracted as the initial sample. For

example, 328 tropical cyclones are recorded at location #1; thus, 328 peak significant wave heights during these

10  tropical cyclones are extracted as the initial sample.

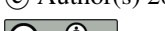

## 4.2 Sensitivity of return values to thresholds

The threshold plays a crucial role in the POT method, which is used to extract the high peak significant wave heights as the sample from the initial sample. When the threshold is suitable, the number of samples is sufficiently large, and the value of the sample is sufficiently high. Based on this sample, the extreme significant

wave height can be extrapolated, and the return significant wave height is reliable. Shao et al., (2018a) and Liang et al., (2019) analysed the sensitivity of the return significant wave height to the threshold. As shown by the theories of the GPD model (Eqs. (2) and (4)), the return significant wave height for a specific return period is dependent on the threshold and the sample (the number and value of samples). When the return significant wave height is stable against the threshold, the sample is stable for extrapolation. Based on these theories and the

influence of the excluded sample on the return significant wave height, Shao et al., (2018a) and Liang et al., (2019) suggested that the suitable threshold should be determined within the stable threshold range.

In the present work, the candidate thresholds within the stable threshold range are further studied to select the unique threshold without a subjective definition. These equally spaced with increasing candidate thresholds are identified by a threshold interval of 0.05 m, which is recommended by Shao et al., (2018a) and Liang et al.,

(2019). For each candidate threshold, the GPD is fitted by using the maximum likelihood estimation method, and the 50-year, 100-year, 150-year and 200-year return significant wave heights are extrapolated. By analysing the return significant wave height, the stable threshold range is uniquely obtained for a specific return period. For example, the stable threshold ranges for the 50-year, 100-year, 150-year and 200-year return periods at location #1 are (3.3 m, 5.75 m), (3.3 m, 5.25 m), (3.3 m, 4.6 m) and (3.3 m, 4.5 m), respectively. The detail on the stable

threshold range can be found in the papers of Shao et al., (2018a) and Liang et al., (2019).



## 5. Characteristics of tropical cyclone waves

In this study, extreme significant wave heights are extrapolated based on tropical cyclone waves. To select the unique threshold without a subjective definition, characteristics of tropical cyclone waves are investigated. The track and intensity of tropical cyclones affect the tropical cyclone wave at the study site. When the track of the tropical cyclone is close to the study site and the intensity of the tropical cyclone is high, the corresponding tropical cyclone wave is sufficiently high for representing the extreme wave at the study site. In this case, the peak significant wave height of this tropical cyclone wave should be extracted as the sample. For example, the peak significant wave height during tropical cyclone Pabuk in 2007 recorded at location #1 is 5.27 m; the peak significant wave height during tropical cyclone Linfa in 2009 recorded at location #1 is 8.17 m; the peak significant wave height during tropical cyclone Molave in 2009 recorded at location #1 is 9.48 m; and the peak significant wave height during tropical cyclone Meranti in 2010 recorded at location #1 is 4.51 m. Tracks of these tropical cyclones are close to location #1 and intensities of these tropical cyclones are high when these tropical cyclones influence waves at location #1 (shown in Fig. 1). In contrast, when the track of the tropical cyclone is far from the study site or the intensity of the tropical cyclone is low, the corresponding tropical cyclone wave is insufficiently high for representing the extreme wave at the study site. In this case, the peak significant wave height of this tropical cyclone wave should not be extracted as the sample. For example, the peak significant wave height during tropical cyclone Maria in 2000 recorded at location #1 is 2.59 m, and the peak significant wave height during tropical cyclone Toraji in 2001 recorded at location #1 is 1.57 m. Although intensities of these tropical cyclones are high when these tropical cyclones influence waves at location #1, tracks of these tropical cyclones are too far from location #1 (shown in Fig. 2). The peak significant wave height during tropical cyclone Trami in 2001 recorded at location #1 is 2.47 m, and the peak significant wave height during tropical




cyclone Wutip in 2007 recorded at location #1 is 2.20 m. Although tracks of these tropical cyclones are close to location #1, intensities of these tropical cyclones are low when these tropical cyclones influence waves at location #1 (shown in Fig. 3). The peak significant wave height during tropical cyclone Kai-tak in 2005 recorded at location #1 is 1.11 m, and the peak significant wave height during tropical cyclone Kammuri in 2008 recorded

5    at location #1 is 2.36 m. Tracks of these tropical cyclones are far from location #1, and intensities of these tropical cyclones are low when these tropical cyclones influence waves at location #1 (shown in Fig. 4).

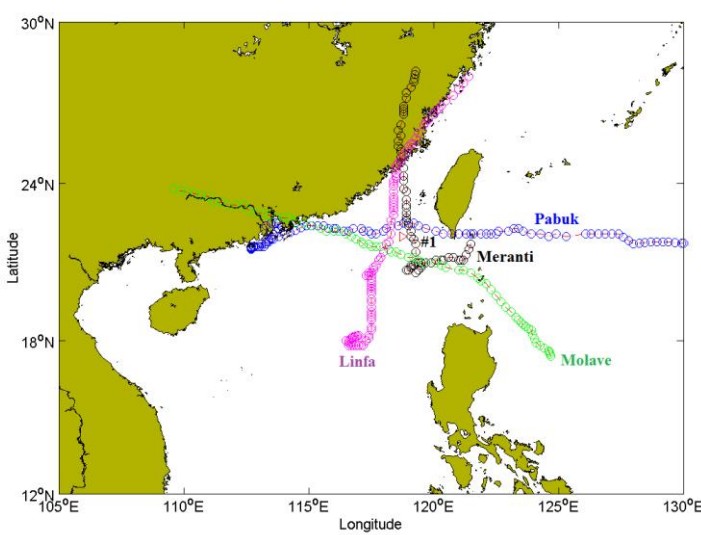

Fig. 1. Tracks of centres of tropical cyclones Pabuk, Linfa, Molave and Meranti (triangle stands for location #1, curves stand for tracks of centres and circles stand for locations of centres).





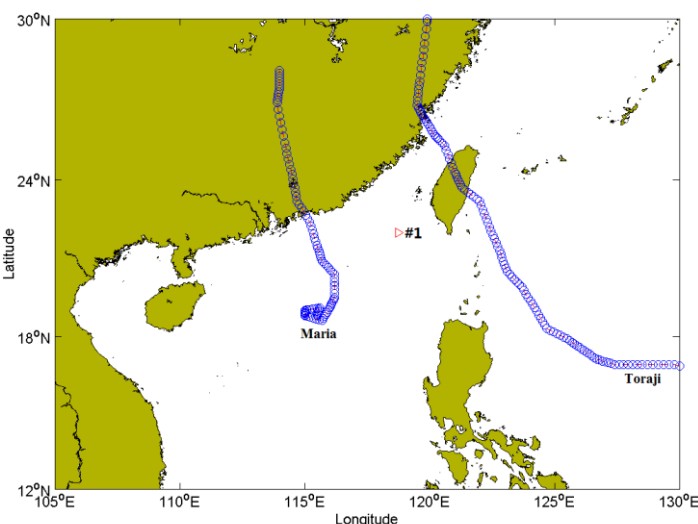

Fig. 2. Tracks of centres of tropical cyclones Maria and Toraji (triangle stands for location #1, curves stand for tracks of centres and circles stand for locations of centres).

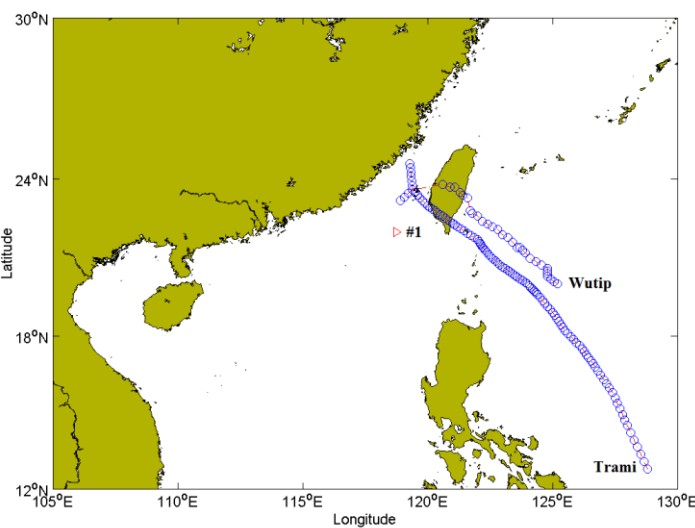

5      Fig. 3. Tracks of centres of tropical cyclones Trami and Wutip (triangle stands for location #1, curves stand for tracks of centres and circles stand for locations of centres).



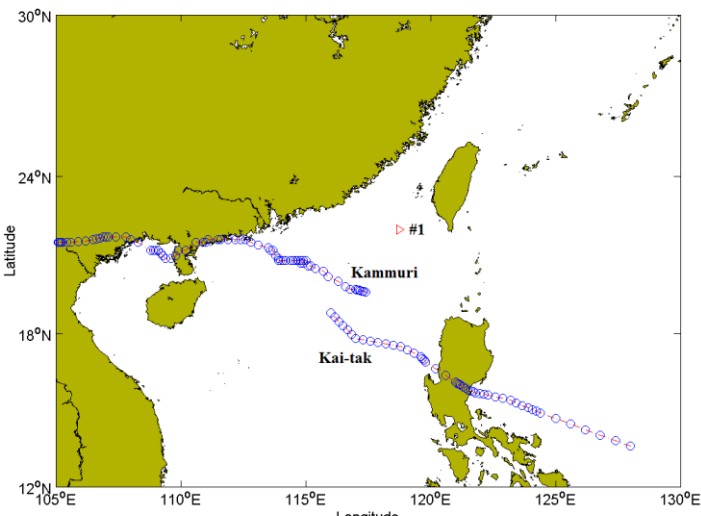

Fig. 4. Tracks of centres of tropical cyclones Kai-tak and Kammuri (triangle stands for location #1, curves stand for tracks of centres and circles stand for locations of centres).

The above analyses show that the track and intensity of tropical cyclones influence the tropical cyclone wave at

5 the targeted location. This influence can be reflected in the distribution of the initial sample (i.e., the distribution of the peak significant wave height). In Fig. 5, the distribution of the initial sample at location #1 is presented. The peak significant wave height is counted from 0 m to 15 m with the interval of 0.05 m (this interval is equal to the threshold interval). It can easily be observed that peak significant wave heights are concentrated in two ranges: range 1 (0-4.15 m) and range 2 (4.15-15 m), with a separation value of 4.15 m. To clearly show ranges 1 and 2,

10 the curve of distribution of peak significant wave heights is plotted. In range 1, 191 peak significant wave heights are found. These peak significant wave heights come from 191 independent tropical cyclone waves, and the corresponding tropical cyclone has a weak influence on the wave at location #1. The track and intensity of these tropical cyclones are analysed, which are similar to those shown in Figs. 2, 3 and 4. In range 2, 137 peak significant wave heights are found. These peak significant wave heights come from 137 independent tropical





cyclone waves, and the corresponding tropical cyclone has a strong influence on the wave at location #1. The track and intensity of these tropical cyclones are analysed, which are similar to those shown in Fig. 1. It can be concluded that the distribution of the initial sample has a natural separation distinguishing the high peak significant wave height from the low peak significant wave height. Moreover, this separation value (the

5 corresponding annual mean number of samples is 3.425) is within the stable threshold range shown in subsection 4.2. Based on the conclusions of Shao et al., (2018a) and Liang et al., (2019), the separation value can be used to extract a stable sample. To further validate the separation value for sampling, the asymptotic tail approximation and estimation uncertainty are analysed. In Fig. 6, the quantile plot for the threshold of 4.15 m is presented, which shows that there are generally few differences between the empirical and fitted values via the GPD model,

indicating a good fit for the selected threshold. In Table 2, the return significant wave height with the confidence interval at location #1 under the threshold of 4.15 m is shown. The return significant wave heights for the return periods of 50-year, 100-year, 150-year and 200-year are 12.07 m, 12.70 m, 13.00 m and 13.20 m, respectively. The likelihood method (Schendel and Thongwichian, 2017) reparametrizes the likelihood in terms of the unknown quantile and uses profile likelihood arguments to construct an approximate 95% confidence interval.

The confidence intervals for the return periods of 50-year, 100-year, 150-year and 200-year are (11.39 m, 13.08 m), (12.02 m, 13.92 m), (12.31 m, 14.36 m) and (12.50 m, 14.66 m), respectively. Their performances indicate that the variance in the extrapolated significant wave heights is acceptable.





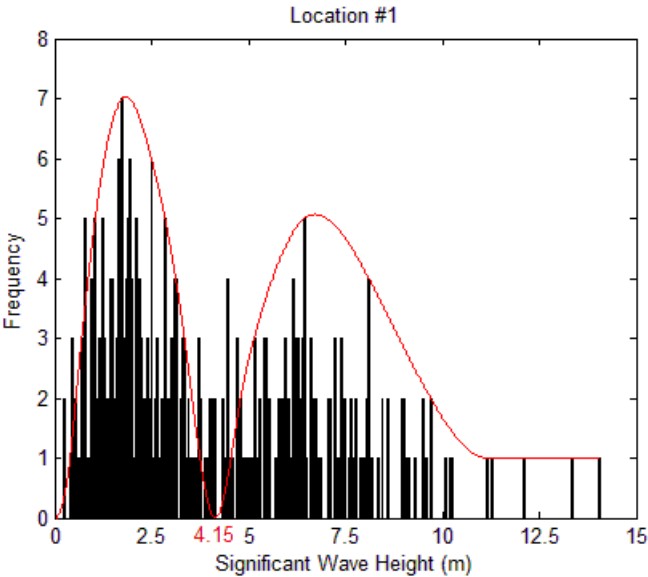

Fig. 5. Histogram of the peak significant wave height from 0 m to 15 m with intervals of 0.05 m at location #1.

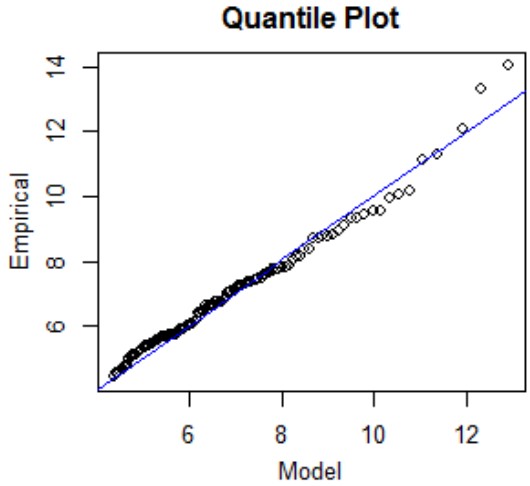

Fig. 6. The quantile plot for GPD-fitted peak significant wave heights at location #1 for the threshold of 4.15 m.

Table 2

Statistics for return significant wave heights and confidence intervals at location #1.

| Return Period | Return Significant Wave Height (m) | Confidence Interval (m) | Width of Confidence Interval (m) |
|---|---|---|---|
| 50-year | 12.07 | (11.39, 13.08) | 1.69 |
| 100-year | 12.70 | (12.02, 13.92) | 1.90 |
| 150-year | 13.00 | (12.31, 14.36) | 2.05 |
| 200-year | 13.20 | (12.50, 14.66) | 2.16 |

The same conclusion can be reached at the other 21 study locations. The separation value shown in the

distribution of the initial sample can be selected as a suitable threshold for sampling. For example, the

distributions of the initial sample for locations #7 and #10 are presented in Fig. 7. The separation values of 3.35

5       m and 4.1 m are selected as the suitable thresholds at locations #7 and #10, respectively. Based on the extracted

sample, the GPD model is used to extrapolate return significant wave heights for different return periods. In

Table 3, the return significant wave heights for the return periods of 50-year, 100-year, 150-year and 200-year are

presented for the other 21 study locations. To validate the reliabilities of the selected threshold and corresponding

return significant wave height, the asymptotic tail approximation and estimation uncertainty are analysed. For

example, the quantile plots for the thresholds of 3.35 m and 4.1 m at locations #7 and #10 are presented in Fig. 8,

and the confidence intervals for the return periods of 50-year, 100-year, 150-year and 200-year at the other 21

study locations are shown in Table 3. The fit results are good and the uncertainties of the return significant wave

height are acceptable.

Based on the threshold selection criterion of Shao et al., (2018a), the highest threshold within the common stable

threshold range is also selected as the suitable threshold for sampling. For example, the threshold of 4.5 m is

obtained at location #1, and the return significant wave heights for the return periods of 50-year, 100-year,

150-year and 200-year are 12.03 m, 12.68 m, 13.00 m and 13.21 m, respectively. At this location, the difference

of the return significant wave height within the stable threshold range is very small. However, some return

significant wave heights within the stable threshold range may be relatively different from the others. For

example, the threshold of 4.25 m is obtained at location #12, and the return significant wave heights for the return

periods of 50-year, 100-year, 150-year and 200-year are 9.59 m, 9.86 m, 9.99 m and 10.06 m, respectively. The

difference of the return significant wave height within the stable threshold range may be relatively large, especially for a short return period. In addition, the return significant wave heights shown in Tables 2 and 3 are compared with the return significant wave heights presented in the paper of Liang et al., (2019). In the present paper, the study locations are same as the study locations presented in the paper of Liang et al., (2019). It can be

found that these two groups of return significant wave heights are similar due to the diagnosis of the return significant wave height within the stable threshold range. If some return significant wave heights within the stable threshold range are relatively different from the others, the corresponding candidate thresholds are rejected. For example, the thresholds of 5.28 m, 4.64 m, 4.2 m and 4.24 m are obtained at location #12 for the return periods of 50-year, 100-year, 150-year and 200-year, respectively. The return significant wave heights for the

return periods of 50-year, 100-year, 150-year and 200-year are 9.69 m, 9.89 m, 9.96 m and 10.05 m, respectively. These four return significant wave heights are similar to return significant wave heights shown in Table 3.

Consequently, threshold selection criteria of Shao et al., (2018a) and Liang et al., (2019) are based on the sensitivity of the return significant wave height to the threshold. These two criteria can be used to assess the extreme significant wave height in any areas in theory. The distribution of the initial sample is based on the

characteristic of the tropical cyclone wave. This criterion may only be used to assess the extreme significant wave height in a tropical cyclone wave-dominated area. However, the distribution of the initial sample can be used to visually distinguish the high peak significant wave height from the low peak significant wave height, and the suitable threshold can uniquely be selected without a subjective definition.



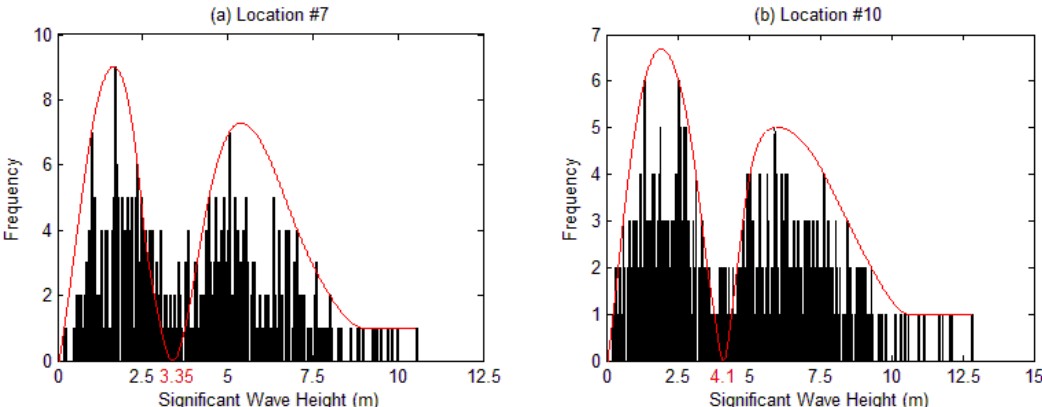

Fig. 7. Histograms of the peak significant wave height at locations #7 and #10.

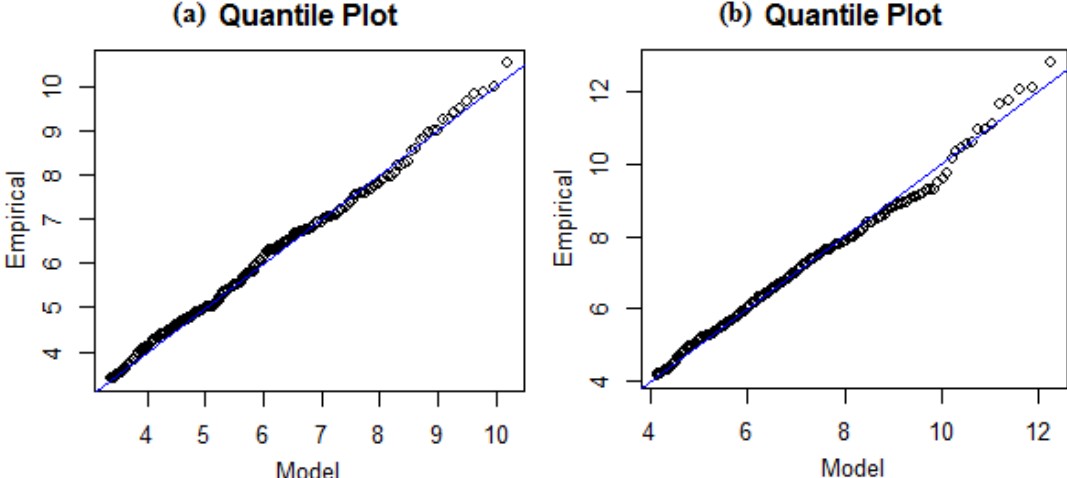

Fig. 8. Quantile plots for GPD-fitted peak significant wave heights ((a) for the threshold of 3.35 m at location #7 and (b) for the threshold of 4.1 m at location #10).



Table 3
Statistics for thresholds, samples and return significant wave heights with 95% confidence intervals.

| Location | Threshold (m) | Annual Mean Number of Samples | Return Significant Wave Heights (m) with 95% Confidence Intervals | | | |
|---|---|---|---|---|---|---|
| | | | 50-year | 100-year | 150-year | 200-year |
| #2 | 3.05 | 3.475 | 9.25 (8.88, 9.88) | 9.58 (9.19, 10.45) | 9.74 (9.37, 10.74) | 9.86 (9.46, 10.92) |
| #3 | 5 | 3.375 | 11.17 (10.74, 11.89) | 11.45 (11.02, 12.34) | 11.61 (11.18, 12.60) | 11.71 (11.29, 12.78) |
| #4 | 4.85 | 4.175 | 12.00 (11.25, 12.91) | 12.24 (11.65, 13.26) | 12.52 (11.93, 13.66) | 12.70 (12.09, 13.94) |
| #5 | 4.95 | 3.975 | 11.84 (11.17, 12.80) | 12.53 (11.81, 13.74) | 12.83 (12.13, 14.25) | 13.06 (12.34, 14.61) |
| #6 | 4.5 | 3.625 | 10.16 (9.92, 10.64) | 10.45 (10.19, 11.01) | 10.56 (10.32, 11.19) | 10.65 (10.39, 11.30) |
| #7 | 3.35 | 5.325 | 9.68 (9.39, 10.11) | 9.96 (9.66, 10.50) | 10.10 (9.82, 10.71) | 10.19 (9.90, 10.84) |
| #8 | 3.6 | 5.55 | 10.36 (10.05, 10.84) | 10.64 (10.26, 11.18) | 10.72 (10.35, 11.32) | 10.91 (10.51, 11.48) |
| #9 | 3.75 | 5.5 | 10.86 (10.49, 11.39) | 11.14 (10.79, 11.82) | 11.28 (10.93, 12.04) | 11.34 (10.98, 12.14) |
| #10 | 4.1 | 5.3 | 11.40 (10.90, 12.04) | 11.87 (11.41, 12.68) | 12.11 (11.58, 13.03) | 12.26 (11.78, 13.23) |
| #11 | 4.25 | 4.75 | 11.44 (11.11, 12.03) | 11.88 (11.56, 12.59) | 12.14 (11.78, 12.87) | 12.29 (11.92, 13.05) |
| #12 | 3.7 | 3.675 | 9.69 (9.37, 10.24) | 9.89 (9.56, 10.57) | 9.93 (9.67, 10.67) | 10.02 (9.76, 10.80) |
| #13 | 3.65 | 5.025 | 11.10 (10.48, 12.07) | 11.63 (10.93, 12.88) | 11.88 (11.15, 13.30) | 12.11 (11.35, 13.68) |
| #14 | 4.15 | 4.8 | 11.06 (10.65, 11.70) | 11.40 (10.99, 12.18) | 11.54 (11.14, 12.41) | 11.66 (11.26, 12.59) |
| #15 | 4.85 | 4.2 | 11.31 (10.92, 11.90) | 11.74 (11.34, 12.44) | 11.95 (11.54, 12.71) | 12.07 (11.67, 12.89) |
| #16 | 4.45 | 3.825 | 10.91 (10.74, 11.38) | 11.31 (11.14, 11.83) | 11.46 (11.28, 12.02) | 11.75 (11.56, 12.33) |
| #17 | 3.05 | 4.775 | 10.31 (9.65, 11.59) | 10.88 (10.03, 12.65) | 11.08 (10.18, 13.15) | 11.26 (10.35, 13.57) |
| #18 | 3.65 | 4.25 | 11.63 (11.04, 12.65) | 12.00 (11.38, 13.30) | 12.18 (11.53, 13.66) | 12.36 (11.70, 13.95) |
| #19 | 3.55 | 2.275 | 7.87 (7.65, 8.33) | 8.16 (7.93, 8.70) | 8.21 (8.00, 8.83) | 8.28 (8.06, 8.91) |
| #20 | 3.65 | 3.575 | 10.07 (9.53, 11.02) | 10.50 (9.94, 11.71) | 10.64 (10.05, 12.02) | 10.84 (10.23, 12.35) |
| #21 | 2.9 | 4 | 10.10 (9.32, 11.59) | 10.70 (9.87, 12.83) | 10.96 (9.94, 13.37) | 11.12 (10.21, 13.99) |
| #22 | 3 | 2.9 | 9.10 (8.57, 10.29) | 9.45 (8.87, 11.01) | 9.58 (9.00, 11.37) | 9.71 (9.09, 11.68) |

## 6. Conclusions

5   In this study, extreme significant wave heights are assessed in the SCS. Before implementing this assessment, the

meteorological phenomenon is analysed to identify the extreme weather. In the SCS, the tropical cyclone

frequently occurs and always drives the storm wave. Thus, the extreme wave is studied based on the tropical





cyclone wave, and a 40-year hindcasted significant wave height of tropical cyclone waves is employed as the initial database. Because this initial database is only simulated during the tropical cyclone and comes from independent tropical cyclone waves, the peak significant wave height of every tropical cyclone wave is directly extracted as the initial sample. The independent and identically distributed characteristic of the initial sample is

satisfied, and the interference of homogenous and declustering methodologies for the selection of the sample is avoided.

Based on the initial sample, the POT method is used to extract the peak significant wave heights over the threshold as the sample. To avoid the subjective definition of the threshold selection criterion, characteristics of tropical cyclone waves are analysed. The analysis results show that the track and intensity of tropical cyclones

affect the sample at the targeted location. When the track of the tropical cyclone is close to the targeted location and the intensity of the tropical cyclone is high, the peak significant wave height of this tropical cyclone wave should be extracted as the sample at the targeted location. In contrast, when the track of the tropical cyclone is far from the targeted location or the intensity of the tropical cyclone is low, the peak significant wave height of this tropical cyclone wave should not be extracted as the sample at the targeted location. These characteristics can be

reflected in the distribution of the initial sample. The separation value is easily observed in the distribution of the initial sample, and this separation value divides the initial sample into the low peak significant wave heights (the corresponding track is far or the corresponding intensity is low) and the high peak significant wave heights (the corresponding track is close and the corresponding intensity is high). Considering that this separation value is within the stable threshold range, this separation value can be used to extract a stable sample for extrapolation.

Therefore, the separation value shown in the distribution of the initial sample is selected as a suitable threshold for sampling. Based on the extracted sample, the GPD model is used to extrapolate the 50-year, 100-year, 150-year and 200-year return significant wave heights at the 22 study locations in the SCS. To validate the

reliabilities of the selected threshold and corresponding return significant wave height, the asymptotic tail approximation and estimation uncertainty are analysed, which show that the selected threshold is suitable and the return significant wave height is reasonable. Considering that the separation value shown in the distribution of the initial sample reflects the characteristic of the tropical cyclone wave, this separation value is suggested for

sampling when an assessment of extreme significant wave heights is needed in a tropical cyclone wave-dominated area (such as the SCS).

**Acknowledgments**

The authors would like to acknowledge the support of the National Science Fund (Grant No. 51679223, 51739010), the 111 Project (No. B14028) , Shandong Provincial Natural Science Key Basic Program

(Grant No.:ZR2017ZA0202) and a grant of the 7th Generation Ultra-Deep-water Drilling Rig Innovation Project.

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
