# Peer review of "Extreme significant wave height of tropical cyclone waves in the South China Sea"

_Natural Hazards and Earth System Sciences, 2018_

## Referee Comment (RC1) · Anonymous Referee #1 · 14 Mar 2019

The manuscript "study of the threshold for the POT method ..." is scientifically interesting, in that it is explores methodologyies for extreme value analysis in presence of tropical cyclones, where the choice of too low thresholds can lead to excessively broad extreme tails, and to irreasonably high return values for high return periods.

However the quality of the manuscript could be improved a lot. First of all, the way the threshold is selected is unclear, and I don't understand many details of what the authors do. Sometimes the text is exceedingly technical, verbose and full of repetitions. Other times relevant aspects of the research are omitted or unclear. Furthermore, the quality of the English is far from optimal.

[Figure]

I would therefore suggest a careful review, that should substantially change the way the methodology is exposed. I would also suggest to have the manuscript copy-read by a motherlanguage English.

Below a list of more detailed comments.

- the title is long and too technical. One has to read it a couple of times to understand the argument. Why not something simpler, like "extreme value analysis of tropical cyclone waves in the Southern Chinese Sea" ?

- lines 14-20 of the abstract: the authors could simply write "A 40-year (1975-2014) hindcast of tropical cyclone waves is used to study the extreme wave heights, employing a Generalized Pareto Distribution (GPD) approach". The rest are details that could be left in the discussion.

- end of the abstract, "initial sample": as far as I can understand, here and elsewhere for the authors the "initial sample" is what I would call the "sample", and for the authors the "sample" is what I would call the "peaks over threshold".

- page 4, line 18: in what way the return levels in AM are unreasonable? And why?

- page 5, line 4: .. which shows that it is possible to study the extreme significant wave height of tropical cyclones

- page 5, lines 5-10: the meaning of these lines is rather unclear.

- page 5, line 7: peak significant wave height, maybe the "maximum significant wave height" would be better, as it is in both space and time.

- page 5, lines 18-19: the words "threshold selection method" are repeated twice in the same sentence (the authors here and elsewhere should avoid so many repetitions)

- page 5, line 20, the acronym ATSME is not introduced.

- page 6, lines 3-5: this is not necessarily true: one could use some automatic technique to decide when the r.l. is not changing.

- page 6, line 10: "the subjective definition still exists in the atsme". It is not entirely clear why

- page 6, lines 17-18: "in section 4 the sampling method is described".

- page 6, line 19: "section 5 discusses the characteristics of ..."

- section 2: this is be the right place to summarize the technique used by the authors to estimate the threshold.

- page 9, line 2: from "nine-hundred" to "tropical cyclone" there is something wrong in this sentence

- section 3.2, a figure with the position of the 22 locations would be useful

- page 10, line 10 and elsewhere, for me this is the sample. The values beyond the threshold are the peaks over threshold

- page 11, line 4, I would add here that the extrapolation here is done fitting the peaks over threshold with a GPD.

- page 11, line 5, you extrapolate only for high return periods, correct?

- page 11, lines 15-20. What technique did you use? ATSME, then the method should be better summarized somewhere, e.g. in section 2.

- page 11, line 19: using ATSME the threshold range depends on the return period? This should be also explained in section 2, and how you can choose a single threshold (I guess, you can take the lowest of the upper limits of the ranges?).

- page 12, line 7, peak -> maximum

- section 5. This (lengthy) discussion does not entirely explain the (interesting) bimodal shape showed by the sample. Is it possible that the 2 modes correspond to 2 different physical characteristics of the TC in this area? Do you have this shape everywhere, or

only in a few locations?

- page 16, line 1, remove waves

- page 16, line 11 "at location 1 above the threshold"

- page 16, lines 12-17, remove the list of values, as they are already in table 2

- page 17, the meaning of figure 6 is not entirely clear. What do the author mean with the word "empirical"?

- section 5: in the end it is not entirely clear how the authors selected the threshold. They used ATSME to select a range for each r.p., and then how did they choose a single threshold? Is it simply the separation point between the modes of the distribution? But that would not be general, as not all the distribution of TC extremes are bimodal.

- section 5: maybe a map with the 22 locations reporting, for example, the 100-year return level would be useful and informative.

- the conclusion could be a little shorter and less technical.

- especially in the conclusion there are several error on English, that should be corrected (the conclusion is a key part of the manuscript).

- line 15: is this 2-modal distribution a consequence of the sampling technique, or is it general, with a physical explanation?
* * *

---

## Referee Comment (RC2) · Anonymous Referee #2 · 2 May 2019

The manuscript "Study of the threshold for the POT method. . ." describes the evaluation of statistical methods to ascertain extreme value wave heights relating to tropical cyclone waves in the South China Sea. The paper is focused on the Peaks Over Threshold method, and accurately defining the cut-off (lower bound) for the extreme value wave heights. The paper is generally well written and well researched in terms of contextualizing the study in relation to existing relevant work.

However, as a general comment I would say this is a very dense paper, which focuses on a particularly specific topic. The paper has some repetition and would benefit from being thinned out considerably so the core relevance, results and impacts of the work

are made clearer. Some paragraphs are very long and could be split down by a factor of two or three. To this end, the paper would benefit from at least one figure up front to break up the text and provide some background on the geographic area.

The overall concept may be of general interest to readers of NHESS but the level of detail is not, and in some places it reads too much like a report than a research paper. The "so what" of the research needs to be better addressed in terms of applications to other work.

I feel the analysis is over-complicated somewhat. Have the authors given any thought to the notion that defining the lower bound wave height for the POT analysis may be as simple as locating where the wave height distribution deviates from a Rayleigh-type to some form of Generalized Extreme Value Distribution? If this is not the case, why not?

Can the authors provide a better definition for "when the tropical cyclone is close to the coast" and "when the intensity is high". Were there distance and intensity thresholds used for this analysis?

---

## Referee Comment (RC3) · Anonymous Referee #3 · 6 May 2019

The authors perform wave climate of South China Sea making usage of 40 years of wave hindcast data. The analysis is accomplished by a discussion on the statistical performance exhibited by a POT model at varying threshold.

The issue addressed by the authors is interesting, current and in line with the topics of the journal. However, here are some methodological aspects that deserve to be better presented and discussed I) Did the authors apply a temporal lag for declustering? II) It is not clear how they generalize the influence of trajectories by means of local conditions (which are not only function of the variable, but also of bathymetry/topography, diffraction and shoaling effects, ...). Please clarify. III) How do they mean with "stable

threshold"? IV) Since every change on bin range of histogram implies changes on related minimum point, how do you set this parameter? Finally, the English language is unsuitable and a deep review with a native speaker is strongly recommended.

---

## Author Comment (AC1) · 4 Jun 2019

Replies to comments by referee #3 Thank you for your comments on our manuscript entitled "Extreme significant wave height of tropical cyclone waves in the South China Sea" (Ref: nhess-2018-349). These comments are all valuable and very helpful for improving our paper. We appreciate that we have a chance to revise the manuscript as you suggested and to resubmit our manuscript after addressing all comments point by point. We hope that the improved manuscript will meet your approval. The main corrections in the manuscript and responses to comments are shown as follows: General response: Thank you for your evaluation of our topic and research. As suggested, we

have rephrased some contents in the manuscript to show the methodological aspects more clearly and strictly. (1) Response: In this study, a 40-year hindcast of tropical cyclone waves is employed as the initial database. The wind used to drive the wave is the blended wind, which covers the entire tropical cyclone process (not only the strong intensity process). The maximal significant wave height of the tropical cyclone wave is obtained during the simulation period. Thus, the maximal significant wave height can be directly extracted as the sample. (2) Response: In this manuscript, we study the extreme significant wave height in the tropical cyclone. When the tropical cyclone track is close to the study site and the tropical cyclone intensity is strong, the wind near the study site is very strong, which primarily determines an extreme wave at the study site. Thus, the track and intensity can be used to analyse the extreme wave at the study site. Location conditions (such as bathymetry/topography, diffraction and shoaling effects) are too complex to be introduced in an extreme value analysis; however, these conditions have been reflected in the tropical cyclone wave simulation. (3) Response: As suggested, "stable threshold" was explained in the manuscript. In the sensitivity of the return significant wave height, when the return significant wave height is stable against an increasing threshold, the corresponding range of candidate thresholds is known as the stable threshold range. See the manuscript P. 4, lines 13-15: "The researchers found that the suitable threshold should be determined within the stable threshold range (i.e., a threshold range corresponding to a range of stable return significant wave heights)." (4) Response: As mentioned, the bin range plays a significant role in the sample distribution. In this study, this range is equal to the threshold interval ($\Delta u=(um-u1)/Ntot$) defined by Liang et al. (2019). u1 is set as the minimal sample, um is set as the maximal sample, and Ntot is set as the number of samples. On the one hand, the sample distribution can be discussed with the sensitivity of the return significant wave height. On the other hand, this definition of the bin range (i.e., the mean interval of the sample) can reflect the sample characteristics in the distribution. See the manuscript P. 8, lines 1-3: "Candidate threshold. Identify the suitable range for the equally spaced and increasing candidate thresholds, (u1, um), and the threshold

interval, Δu=(um-u1)/Ntot. u1 is set as the minimal sample, um is set as the maximal sample, and Ntot is set as the number of samples." See the manuscript P. 15, lines 1-2: "The sample is counted from 0 m to 15 m with an interval of 0.05 m, which is the same as the threshold interval." (5) Response: As suggested, we have carefully reviewed the manuscript. To further improve the quality, proofreading and language editing have been completed by American Journal Experts.

---

## Author Response (AR1)

**Replies to comments by referees**

**Extreme significant wave height of tropical cyclone waves in the South China Sea**

Zhuxiao Shao[1], Bingchen Liang[1, 2], Huajun Li[1, 2], Ping Li[3], Dongyoung Lee[1, 4]

[1] College of Engineering, Ocean University of China, 238 Songling Road, Qingdao 266100, China

[2] Shandong Province Key Laboratory of Ocean Engineering, Ocean University of China, 238 Songling Road, Qingdao 266100, China

[3] China Classification Society, Beijing 100007, China

[4] Korea Institute of Ocean Science & Technology, Ansan, Korea

Thank you for your comments on our manuscript entitled "Extreme significant wave height of tropical cyclone waves in the South China Sea" (Ref: nhess-2018-349). These comments are all valuable and very helpful for improving our paper. We appreciate that we have a chance to revise the manuscript as you suggested and to resubmit our manuscript after addressing all comments point by point. We hope that the improved manuscript will meet your approval.

The main corrections in the manuscript and responses to comments are shown as follows:

**Referee #1**

The manuscript "study of the threshold for the POT method ..." is scientifically interesting, in that it is explores methodologyies for extreme value analysis in presence of tropical cyclones, where the choice of too low thresholds can lead to excessively broad extreme tails, and to irreasonably high return values for high return periods.

However the quality of the manuscript could be improved a lot. First of all, the way the threshold is selected is unclear, and I don't understand many details of what the authors do. Sometimes the text is exceedingly technical, verbose and full of repetitions. Other times relevant aspects of the research are omitted or unclear. Furthermore, the quality of the English is far from optimal.

I would therefore suggest a careful review, that should substantially change the way the methodology is exposed. I would also suggest to have the manuscript copy-read by a motherlanguage English.

> **Response:** Thank you for your evaluation of our manuscript. As suggested, some of the contents in the manuscript have been rewritten to improve the quality of the manuscript, and the threshold selection method is described clearly in detail. In addition, we have rephrased the paper to present ideas more concisely and strictly. To further improve the manuscript, proofreading and language editing have been completed by American Journal Experts.

[Figure]

**EDITORIAL CERTIFICATE**

This document certifies that the manuscript listed below was edited for proper English language, grammar, punctuation, spelling, and overall style by one or more of the highly qualified native English speaking editors at American Journal Experts.

**Manuscript title:**

Extreme significant wave height of tropical cyclone waves in the South China Sea

**Authors:**

Zhuxiao Shao, Bingchen Liang, Huajun Li, Ping Li, Dongyoung Lee

**Date Issued:**

June 3, 2019

**Certificate Verification Key:**

548C-8CC5-581E-899E-E34P

[Figure]

This certificate may be verified at www.aje.com/certificate. This document certifies that the manuscript listed above was edited for proper English language, grammar, punctuation, spelling, and overall style by one or more of the highly qualified native English speaking editors at American Journal Experts. Neither the research content nor the authors' intentions were altered in any way during the editing process. Documents receiving this certification should be English-ready for publication; however, the author has the ability to accept or reject our suggestions and changes. To verify the final AJE edited version, please visit our verification page. If you have any questions or concerns about this edited document, please contact American Journal Experts at support@aje.com.

American Journal Experts provides a range of editing, translation and manuscript services for researchers and publishers around the world. Our top-quality PhD editors are all native English speakers from America's top universities. Our editors come from nearly every research field and possess the highest qualifications to edit research manuscripts written by non-native English speakers. For more information about our company, services and partner discounts, please visit www.aje.com.

**(1) the title is long and too technical. One has to read it a couple of times to understand the argument. Why not something simpler, like "extreme value analysis of tropical cyclone waves in the Southern Chinese Sea" ?**

**Response:** As suggested, the title has been changed to "Extreme significant wave height of tropical cyclone waves in the South China Sea".

**(2) lines 14-20 of the abstract: the authors could simply write "A 40-year (1975-2014) hindcast of tropical cyclone waves is used to study the extreme wave heights, employing a Generalized Pareto Distribution (GPD) approach". The rest are details that could be left in the discussion.**

**Response:** As suggested, the contents of the corresponding lines have been simplified.

● See the manuscript P. 1, lines 13-15: "In this study, a 40-year (1975-2014) hindcast of tropical cyclone waves is used to analyse the extreme significant wave height,

employing the peak over threshold (POT) method with the generalized Pareto

distribution (GPD) model."

**(3) end of the abstract, "initial sample": as far as I can understand, here and elsewhere for the authors the "initial sample" is what I would call the "sample", and for the authors the "sample" is what I would call the "peaks over threshold".**

**Response:** As suggested, "initial sample" was renamed "sample", and "sample" was renamed "extreme sample" (i.e., peaks over threshold).

**(4) page 4, line 18: in what way the return levels in AM are unreasonable? And why?**

**Response:** As suggested, "the return levels in AM are unreasonable" was clearly described and explained in the manuscript.

- See the manuscript P. 3, lines 19-22; P. 4, lines 1-10: "Shao et al. (2018a) compared

  the annual maxima (AM) method (Tawn, 1988) with the POT method. The AM

  method is an easy sampling method that does not require additional work, as the

  method directly extracts the annual maximal significant wave height for extrapolation.

  However, the AM method has limitations in a fixed time window (i.e., one year),

  which cannot guarantee the independence and number of samples. The annual

  maximal significant wave height obtained from neighbouring years may originate

  from the same extreme wave; some maximal significant wave heights may be

  neglected (i.e., the annual maximal significant wave height may be smaller than some

  unselected maximal significant wave heights in other years), resulting in an

  insufficient number of samples, especially for a relatively long return period. In a

  tropical cyclone, the AM method's limitation is further exacerbated, even if the return

period is close to the database size. The annual frequency, intensity and track of recorded tropical cyclones greatly vary, and corresponding waves have obvious differences. Shao et al. (2018a) found that the minimal sample may be much less than the maximal sample, and the minimal sample may be too small to represent the extreme wave (i.e., the minimal sample in the AM method is obviously smaller than the extreme sample in the POT method)."

**(5) - page 5, line 4: .. which shows that it is possible to study the extreme significant wave height of tropical cyclones**

**Response:** As suggested, the content in the corresponding line was rephrased.

- See the manuscript P. 5, lines 5-6: "Thus, it is possible to study the extreme significant wave height in a tropical cyclone."

**(6) - page 5, lines 5-10: the meaning of these lines is rather unclear.**

**Response:** As suggested, the contents in the corresponding lines were rewritten to clearly show our ideas.

- See the manuscript P. 5, lines 6-9: "To achieve the assessment, a 40-year (1975-2014) hindcasted significant wave height of tropical cyclone waves is employed as the initial database. Considering that the hindcast is independently simulated during the tropical cyclone recorded in the SCS, the maximal significant wave height of the tropical cyclone wave can be directly extracted as the sample when the tropical cyclone influences the wave at the targeted location."

**(7) - page 5, line 7: peak significant wave height, maybe the "maximum significant wave height" would be better, as it is in both space and time.**

**Response:** As suggested, "peak significant wave height" was renamed "maximal significant wave height".

**(8) - page 5, lines 18-19: the words "threshold selection method" are repeated twice in the same sentence (the authors here and elsewhere should avoid so many repetitions)**

**Response:** As suggested, the contents in the corresponding lines were rephrased to avoid repetition. In addition, we rephrased the paper to present our ideas more concisely.

- See the manuscript P. 4, lines 15-18: "Based on this conclusion, Shao et al. (2018a) defined the largest threshold within the common stable threshold range as the suitable threshold, and Liang et al. (2019) proposed an Automated Threshold Selection Method based on the characteristic of Extrapolated significant wave heights (the acronym is ATSME)."

**(9) - page 5, line 20, the acronym ATSME is not introduced.**

**Response:** As suggested, the acronym ATSME was introduced.

- See the manuscript P. 4, lines 15-18: "Based on this conclusion, Shao et al. (2018a) defined the largest threshold within the common stable threshold range as the suitable threshold, and Liang et al. (2019) proposed an Automated Threshold Selection Method based on the characteristic of Extrapolated significant wave heights (the acronym is ATSME)."

**(10) - page 6, lines 3-5: this is not necessarily true: one could use some automatic technique to decide when the r.l. is not changing.**

**Response:** As suggested, the contents in the corresponding lines were deleted.

**(11) - page 6, line 10: "the subjective definition still exists in the atsme". It is not entirely clear why.**

**Response:** In the ATSME, the maximal threshold of the stable threshold range is used to extract the extreme sample. Considering that the selected threshold is within the stable threshold range, the influence of this definition is small for the return significant wave heights. As mentioned, the content in corresponding lines may mislead readers. Thus, we have deleted the corresponding contents.

**(12) - page 6, lines 17-18: "in section 4 the sampling method is described".**

**Response:** As suggested, the contents in the corresponding lines were rewritten.

- See the manuscript P. 5, line 16: "In Section 4, the sampling method is described."

**(13) - page 6, line 19: "section 5 discusses the characteristics of ..."**

**Response:** As suggested, the content in the corresponding line was rewritten.

- See the manuscript P. 5, lines 16-17: "In Section 5, the characteristics of tropical cyclone waves are discussed."

**(14) - section 2: this is be the right place to summarize the technique used by the authors to estimate the threshold.**

**Response:** As suggested, the technique used by the authors was summarized in Section 2.

- See the manuscript P. 6, Background.

**(15) - page 9, line 2: from "nine-hundred" to "tropical cyclone" there is something wrong in this sentence**

**Response:** As suggested, the content in the corresponding line was rewritten.

- See the manuscript P. 9, lines 10-11: "From 1975 to 2014, waves are simulated only

during 974 independent tropical cyclones."

**(16) - section 3.2, a figure with the position of the 22 locations would be useful**

**Response:** As suggested, a figure with the positions of the 22 sample locations is presented in subsection 3.2.

● See the manuscript P. 10, Fig.1.

[Figure]

Fig. 1. The study sites in the study region.

**(17) - page 10, line 10 and elsewhere, for me this is the sample. The values beyond the threshold are the peaks over threshold**

**Response:** As suggested, "initial sample" was renamed "sample", and "sample" was renamed "extreme sample" (i.e., peaks over threshold).

**(18) - page 11, line 4, I would add here that the extrapolation here is done fitting the peaks over threshold with a GPD.**

**Response:** As suggested, the content in the corresponding line was rewritten.

**(19) - page 11, line 5, you extrapolate only for high return periods, correct?**

**Response:** Yes, high return periods are extrapolated. The corresponding content is emphasized in the manuscript.

**(20) - page 11, lines 15-20. What technique did you use? ATSME, then the method should be better summarized somewhere, e.g. in section 2.**

**Response:** As suggested, the ATSME was summarized in subsection 2.2.

● See the manuscript P. 7, subsection 2.2.

**(21) - page 11, line 19: using ATSME the threshold range depends on the return period? This should be also explained in section 2, and how you can choose a single threshold (I guess, you can take the lowest of the upper limits of the ranges?).**

**Response:** The stable threshold range shows a pattern associated with the return period. As suggested, this phenomenon was explained in subsection 2.2. In the ATSME, the suitable threshold is defined as the maximal threshold of the stable threshold range to guarantee design security.

● See the manuscript P. 8, lines 10-11: "Suitable threshold. Determine the suitable threshold within the stable threshold range, such as the maximal threshold."

● See the manuscript P. 8, lines 12-18; P. 9, lines 1-2: "By the ATSME, a unique threshold is determined within a uniquely stable threshold range for a specific return period. Liang et al. (2019) found that the stable threshold range shows a pattern associated with the return period. The minimal threshold of the stable threshold range controls the representativeness of the extreme sample; thus, the samples over the minimal threshold can represent extreme waves well, and the minimal thresholds for

different return periods remain constant. The maximal threshold of the stable

threshold range controls the number of extreme samples, and a longer return period

requires more extreme samples; thus, the maximal thresholds gradually decrease

when the return period increases. Consequently, excluding the sample within the

stable threshold ranges does not obviously influence the return significant wave

heights, and a suitable threshold should be determined within the stable threshold

range."

**(22) page 12, line 7, peak -> maximum**

**Response:** As suggested, "peak significant wave height" was renamed "maximal significant wave height".

**(23) - section 5. This (lengthy) discussion does not entirely explain the (interesting) bimodal shape showed by the sample. Is it possible that the 2 modes correspond to 2 different physical characteristics of the TC in this area? Do you have this shape everywhere, or only in a few locations?**

**Response:** As suggested, we have rephrased the content in Section 5 to present our ideas more concisely. The initial database and characteristics of tropical cyclones determine a bimodal shape. During a tropical cyclone, when the track is close and the intensity is strong, the maximal significant wave height can represent the extreme wave at the targeted location. However, it is difficult to determine the extreme sample through track threshold and intensity threshold. In this study, we use a fixed distance to identify the initial database at the study site. When the distance between the centre of the tropical cyclone and the study site is within 300 km, hourly significant wave heights simulated during this tropical cyclone are adopted as the initial database at the study site. This fixed distance allows some small samples (the corresponding track is far, or the intensity is weak) to be extracted.

Thus, other analyses are needed to identify the extreme sample from the sample, such as the sample distribution with the sensitivity of the return significant wave height.

At the 22 study sites in the SCS, a bimodal shape exists. We think that this bimodal shape is obvious in the tropical cyclone-dominated area when a fixed distance is used. In this area, the tropical cyclone always drives the storm wave, and the number of tropical cyclones is sufficiently large (the annual mean number is greater than 5 (Mazas and Hamm, 2011)).

- See the manuscript P. 19, lines 11-14: "Based on further analysis of this distribution, the initial database and characteristics of the tropical cyclones determine a bimodal shape. A fixed distance is used to identify the initial database at the study site. This fixed distance allows some small samples (the corresponding track is far, or the intensity is weak) to be extracted. Thus, other analyses are needed to identify the extreme sample from the sample."

- See the manuscript P. 19, lines 15-19: "Consequently, the results of this study present a concept linking the assessment of extreme significant wave heights with the characteristics of tropical cyclones in a tropical cyclone-dominated area. The sample at the targeted location is affected by the track and intensity of the tropical cyclone. Future studies are suggested to promote the assessment of extreme significant wave heights in a tropical cyclone. For example, the threshold may be determined directly through a combination of track threshold and intensity threshold."

**(24) - page 16, line 1, remove waves**

**Response:** As suggested, "waves" was removed.

**(25) - page 16, line 11 "at location 1 above the threshold"**

**Response:** As suggested, the content in the corresponding line was rewritten.

**(26) - page 16, lines 12-17, remove the list of values, as they are already in table 2**

**Response:** As suggested, the list of values was removed.

**(27) - page 17, the meaning of figure 6 is not entirely clear. What do the author mean with the word "empirical"?**

**Response:** As suggested, the meaning of Fig. 6 was clearly described in the manuscript. The quantile plot was discussed by Coles (2001) and produced by a free package running in R. The term "empirical" represents "empirical quantile", and "model" represents "model quantile".

- See the manuscript P. 15, lines 14-16: "The asymptotic tail approximation can be estimated by the quantile plot, which is discussed by Coles (2001) and produced by a free package running in R."

**(28) - section 5: in the end it is not entirely clear how the authors selected the threshold. They used ATSME to select a range for each r.p., and then how did they choose a single threshold? Is it simply the separation point between the modes of the distribution? But that would not be general, as not all the distribution of TC extremes are bimodal.**

**Response:** As suggested, the method of threshold selection was clearly described in Section 5. In the ATSME, the maximal threshold of the stable threshold range is used to extract the extreme sample.

In the tropical cyclone, the track and intensity affect the sample at the targeted location. To assess the extreme significant wave height, we use a fixed distance to identify the initial database at the study site. This fixed distance allows some small samples (the corresponding track is far, or the intensity is weak) to be extracted. Thus, other analyses are needed to identify the extreme sample from the sample. In the sample distribution, a

separation distinguishes the high sample from the low sample. In addition, this separation is within the stable threshold range. Thus, this separation can be used to extract the extreme sample.

We think that this method is suitable in the tropical cyclone-dominated area when a fixed distance is used. In this area, the tropical cyclone always drives the storm wave, and the number of tropical cyclones is sufficiently large (the annual mean number is greater than 5 (Mazas and Hamm, 2011)). To guarantee design security, a sensitivity analysis is suggested to supplement the threshold selection in the distribution.

**(29) - section 5: maybe a map with the 22 locations reporting, for example, the 100-year return level would be useful and informative.**

**Response:** As suggested, a map with the 22 sample locations is shown in Fig. 1. In addition, the 100-year return level is presented in Tables 1 and 2.

**(30) - the conclusion could be a little shorter and less technical.**

**Response:** As suggested, we have rephrased the conclusion to present our ideas more concisely and strictly.

● See the manuscript P. 18, conclusions and discussions.

**(31) - especially in the conclusion there are several error on English, that should be corrected (the conclusion is a key part of the manuscript).**

**Response:** As suggested, we have carefully reviewed the conclusion. To further improve it, proofreading and language editing have been completed by American Journal Experts.

**(32) - line 15: is this 2-modal distribution a consequence of the sampling technique, or is it general, with a physical explanation?**

**Response:** The initial database and characteristics of tropical cyclones determine the bimodal shape. A fixed distance is used to identify the initial database at the study site.

This fixed distance allows some small samples (the corresponding track is far, or the intensity is weak) to be extracted. Thus, other analyses are needed to identify the extreme sample from the sample, such as the sample distribution with the sensitivity of the return significant wave height.

**Referee #2**

The manuscript "Study of the threshold for the POT method. . ." describes the evaluation of statistical methods to ascertain extreme value wave heights relating to tropical cyclone waves in the South China Sea. The paper is focused on the Peaks Over Threshold method, and accurately defining the cut-off (lower bound) for the extreme value wave heights. The paper is generally well written and well researched in terms of contextualizing the study in relation to existing relevant work.

However, as a general comment I would say this is a very dense paper, which focuses on a particularly specific topic. The paper has some repetition and would benefit from being thinned out considerably so the core relevance, results and impacts of the work are made clearer. Some paragraphs are very long and could be split down by a factor of two or three. To this end, the paper would benefit from at least one figure up front to break up the text and provide some background on the geographic area.

**Response:** Thank you for your evaluation of our research. As suggested, some contents in the manuscript have been rewritten to show our ideas more concisely. Some repetitive information was deleted to clearly show the core relevance, results and impacts of our work.

In the manuscript, the long paragraph was rephrased and divided into two or three paragraphs. In addition, figures and tables appear in appropriate locations to break up the text.

In Fig. 1, the study region with the 22 study sites is shown. The SCS is the largest and deepest marginal sea in the western Pacific Ocean. In this area, the tropical cyclone always drives the storm wave, and the number of tropical cyclones is sufficiently large. Thus, the extreme significant wave height can be assessed in a tropical cyclone.

To further improve the quality, proofreading and language editing have been completed by American Journal Experts.

[Figure]

**EDITORIAL CERTIFICATE**

This document certifies that the manuscript listed below was edited for proper English language, grammar, punctuation, spelling, and overall style by one or more of the highly qualified native English speaking editors at American Journal Experts.

**Manuscript title:**

Extreme significant wave height of tropical cyclone waves in the South China Sea

**Authors:**

Zhuxiao Shao, Bingchen Liang, Huajun Li, Ping Li, Dongyoung Lee

**Date Issued:**

June 3, 2019

**Certificate Verification Key:**

548C-8CC5-581E-899E-E34P

[Figure]

This certificate may be verified at www.aje.com/certificate. This document certifies that the manuscript listed above was edited for proper English language, grammar, punctuation, spelling, and overall style by one or more of the highly qualified native English speaking editors at American Journal Experts. Neither the research content nor the authors' intentions were altered in any way during the editing process. Documents receiving this certification should be English-ready for publication; however, the author has the ability to accept or reject our suggestions and changes. To verify the final AJE edited version, please visit our verification page. If you have any questions or concerns about this edited document, please contact American Journal Experts at support@aje.com.

American Journal Experts provides a range of editing, translation and manuscript services for researchers and publishers around the world. Our top-quality PhD editors are all native English speakers from America's top universities. Our editors come from nearly every research field and possess the highest qualifications to edit research manuscripts written by non-native English speakers. For more information about our company, services and partner discounts, please visit www.aje.com.

**(1) The overall concept may be of general interest to readers of NHESS but the level of detail is not, and in some places it reads too much like a report than a research paper. The "so what" of the research needs to be better addressed in terms of applications to other work.**

**Response:** As suggested, we have rephrased the manuscript to make it stricter and to present some of the contents in greater detail. We try to show our research more clearly to enable its application to other works.

**(2) I feel the analysis is over-complicated somewhat. Have the authors given any thought to the notion that defining the lower bound wave height for the POT analysis may be as simple as locating where the wave height distribution deviates from a Rayleigh-type to some form of Generalized Extreme Value Distribution? If this is not the case, why not?**

**Response:** As suggested, we have rephrased the paper to present our ideas more concisely.

In the SCS, Shao et al. (2018a) compared the AM method with the POT method. Due to a fixed time window (i.e., one year), the independence and number of samples cannot be guaranteed. In a tropical cyclone, the influence of this fixed time window is further exacerbated, even if the return period is close to the size of the database. Compared with the AM method, the POT method is a natural sampling method without additional limitations. When the threshold is suitable, the POT method can guarantee the representativeness and number of extreme samples. However, the process of threshold selection is relatively complex.

Shao et al. (2018a) and Liang et al. (2019) analysed the sensitivity of the return significant wave height to the threshold. The researchers found that a suitable threshold should be determined within the stable threshold range. However, a unique threshold cannot be directly selected. To determine a unique threshold, Shao et al. (2018a) defined the largest threshold within the common stable threshold range as the suitable threshold, and Liang et al. (2019) proposed the use of an ATSME. The ATSME selects the largest threshold within the stable threshold range as the suitable threshold for different return periods.

In this paper, we further studied the sensitivity with the characteristic of tropical cyclones. We want to present a concept linking the assessment with this characteristic in a tropical cyclone-dominated area. We analysed the track and intensity influences of tropical cyclones on the extreme wave at the targeted location and studied the distribution of the sample with the sensitivity. To validate the high sample in the distribution for extrapolation, we estimated the asymptotic tail approximation and estimation uncertainty. As mentioned, it is interesting to study the location of the distribution deviating from a Rayleigh-type to some GEV forms. Thank you for your suggestion. This concept is significant, and we may research this topic in our future studies.

- See the manuscript P. 3, lines 19-22; P. 4, lines 1-10: "Shao et al. (2018a) compared

the annual maxima (AM) method (Tawn, 1988) with the POT method. The AM method is an easy sampling method that does not require additional work, as the method directly extracts the annual maximal significant wave height for extrapolation. However, the AM method has limitations in a fixed time window (i.e., one year), which cannot guarantee the independence and number of samples. The annual maximal significant wave height obtained from neighbouring years may originate from the same extreme wave; some maximal significant wave heights may be neglected (i.e., the annual maximal significant wave height may be smaller than some unselected maximal significant wave heights in other years), resulting in an insufficient number of samples, especially for a relatively long return period. In a tropical cyclone, the AM method's limitation is further exacerbated, even if the return period is close to the database size. The annual frequency, intensity and track of recorded tropical cyclones greatly vary, and corresponding waves have obvious differences. Shao et al. (2018a) found that the minimal sample may be much less than the maximal sample, and the minimal sample may be too small to represent the extreme wave (i.e., the minimal sample in the AM method is obviously smaller than the extreme sample in the POT method)."

- See the manuscript P. 4, lines 10-21: "Compared with the AM method, the POT method is a natural sampling method without additional limitations. When the threshold is suitable, the POT method can guarantee the representativeness and number of extreme samples. However, the threshold selection process is relatively complex. Shao et al. (2018a) and Liang et al. (2019) analysed the sensitivity of the

return significant wave height to the threshold. The researchers found that the suitable threshold should be determined within the stable threshold range (i.e., a threshold range corresponding to a range of stable return significant wave heights). Based on this conclusion, Shao et al. (2018a) defined the largest threshold within the common stable threshold range as the suitable threshold, and Liang et al. (2019) proposed an Automated Threshold Selection Method based on the characteristic of Extrapolated significant wave heights (the acronym is ATSME). The ATSME employs the differences in extrapolated significant wave heights for neighbouring thresholds as the diagnostic parameters to identify the uniquely stable threshold range via an automated method and selects the largest threshold within the stable threshold range as the suitable threshold for different return periods."

**(3) Can the authors provide a better definition for "when the tropical cyclone is close to the coast" and "when the intensity is high". Were there distance and intensity thresholds used for this analysis?**

**Response:** As suggested, we explained the corresponding content regarding the track and intensity in the manuscript.

The track and intensity of tropical cyclones affect the sample at the targeted location. When the track of the tropical cyclone is close to the study site and the intensity of the tropical cyclone is strong, the corresponding wave is sufficiently strong to represent the extreme wave at the study site. In contrast, when the track is far or the intensity is weak, the corresponding wave is insufficiently strong. However, it is difficult to determine the extreme sample through the track threshold and intensity threshold. A combination of track and intensity is relatively complex. In this study, we use a fixed distance to identify the initial database at the study site. When the distance between the centre of the tropical

cyclone and the study site is within 300 km, hourly significant wave heights simulated during this tropical cyclone are adopted as the initial database at the study site. This fixed distance allows some small samples (the corresponding track is far, or the intensity is weak) to be extracted. Thus, other analyses are needed to identify the extreme sample from the sample, such as the sample distribution with the sensitivity of the return significant wave height. We will continue to study the assessment in the tropical cyclone. We hope that we will discover a combination of track threshold and intensity threshold, or the results of this paper can stimulate more scholars to pay attention to this topic.

- See the manuscript P. 19, lines 15-19: "Consequently, the results of this study present a concept linking the assessment of extreme significant wave heights with the characteristics of tropical cyclones in a tropical cyclone-dominated area. The sample at the targeted location is affected by the track and intensity of the tropical cyclone. Future studies are suggested to promote the assessment of extreme significant wave heights in a tropical cyclone. For example, the threshold may be determined directly through a combination of track threshold and intensity threshold."

**Referee #3**

The authors perform wave climate of South China Sea making usage of 40 years of wave hindcast data. The analysis is accomplished by a discussion on the statistical performance exhibited by a POT model at varying threshold.

The issue addressed by the authors is interesting, current and in line with the topics of the journal. However, here are some methodological aspects that deserve to be better presented and discussed.

**Response:** Thank you for your evaluation of our topic and research. As suggested, we have rephrased some contents in the manuscript to show the methodological aspects more clearly and strictly.

**(1) Did the authors apply a temporal lag for declustering?**

**Response:** In this study, a 40-year hindcast of tropical cyclone waves is employed as the initial database. The wind used to drive the wave is the blended wind, which covers the entire tropical cyclone process (not only the strong intensity process). The maximal significant wave height of the tropical cyclone wave is obtained during the simulation period. Thus, the maximal significant wave height can be directly extracted as the sample.

**(2) It is not clear how they generalize the influence of trajectories by means of local conditions (which are not only function of the variable, but also of bathymetry/topography, diffraction and shoaling effects, ...). Please clarify.**

**Response:** In this manuscript, we study the extreme significant wave height in the tropical cyclone. When the tropical cyclone track is close to the study site and the tropical cyclone intensity is strong, the wind near the study site is very strong, which primarily determines an extreme wave at the study site. Thus, the track and intensity can be used to analyse the extreme wave at the study site. Location conditions (such as bathymetry/topography, diffraction and shoaling effects) are too complex to be introduced in an extreme value

analysis; however, these conditions have been reflected in the tropical cyclone wave simulation.

**(3) How do they mean with "stable threshold"?**

**Response:** As suggested, "stable threshold" was explained in the manuscript. In the sensitivity of the return significant wave height, when the return significant wave height is stable against an increasing threshold, the corresponding range of candidate thresholds is known as the stable threshold range.

- See the manuscript P. 4, lines 13-15: "The researchers found that the suitable threshold should be determined within the stable threshold range (i.e., a threshold range corresponding to a range of stable return significant wave heights)."

**(4) Since every change on bin range of histogram implies changes on related minimum point, how do you set this parameter?**

**Response:** As mentioned, the bin range plays a significant role in the sample distribution. In this study, this range is equal to the threshold interval ($\Delta u = \frac{um - u1}{Ntot}$) defined by Liang et al. (2019). $u_1$ is set as the minimal sample, $u_m$ is set as the maximal sample, and $N_{tot}$ is set as the number of samples. On the one hand, the sample distribution can be discussed with the sensitivity of the return significant wave height. On the other hand, this definition of the bin range (i.e., the mean interval of the sample) can reflect the sample characteristics in the distribution.

- See the manuscript P. 8, lines 1-3: "Candidate threshold. Identify the suitable range for the equally spaced and increasing candidate thresholds, *(u₁, uₘ)*, and the threshold interval, $\Delta u = \frac{um - u1}{Ntot}$. $u_1$ is set as the minimal sample, $u_m$ is set as the maximal sample, and $N_{tot}$ is set as the number of samples."

- See the manuscript P. 15, lines 1-2: "The sample is counted from 0 m to 15 m with an interval of 0.05 m, which is the same as the threshold interval."

**(5) Finally, the English language is unsuitable and a deep review with a native speaker is strongly recommended.**

**Response:** As suggested, we have carefully reviewed the manuscript. To further improve the quality, proofreading and language editing have been completed by American Journal Experts.

[Figure]

**EDITORIAL CERTIFICATE**

This document certifies that the manuscript listed below was edited for proper English language, grammar, punctuation, spelling, and overall style by one or more of the highly qualified native English speaking editors at American Journal Experts.

**Manuscript title:**
Extreme significant wave height of tropical cyclone waves in the South China Sea

**Authors:**
Zhuxiao Shao, Bingchen Liang, Huajun Li, Ping Li, Dongyoung Lee

**Date Issued:**
June 3, 2019

**Certificate Verification Key:**
548C-8CC5-581E-899E-E34P

[Figure]

This certificate may be verified at www.aje.com/certificate. This document certifies that the manuscript listed above was edited for proper English language, grammar, punctuation, spelling, and overall style by one or more of the highly qualified native English speaking editors at American Journal Experts. Neither the research content nor the authors' intentions were altered in any way during the editing process. Documents receiving this certification should be English-ready for publication; however, the author has the ability to accept or reject our suggestions and changes. To verify the final AJE edited version, please visit our verification page. If you have any questions or concerns about this edited document, please contact American Journal Experts at support@aje.com.

American Journal Experts provides a range of editing, translation and manuscript services for researchers and publishers around the world. Our top-quality PhD editors are all native English speakers from America's top universities. Our editors come from nearly every research field and possess the highest qualifications to edit research manuscripts written by non-native English speakers. For more information about our company, services and partner discounts, please visit www.aje.com.

[revised manuscript text omitted]

---

## Editor Decision (ED1)

**Comments from the  Editor**

- Page 18, line 4 replace "Conclusions and discussions" with "Discussion and conclusions"
- I still consider the conclusions of your study not clearly written in the final section (see below)

In my view the three paragraphs of the "Conclusions"  could be reconsidered as the following:

- First paragraph:

I suggest the following rephrasing:

*Estimates of the return significant wave heights within the stable threshold range may be relatively different, depending on the adopted threshold criterion, especially for a short return period. In general, the threshold selection criterion of Shao et al. (2018a) can be used to assess the extreme significant wave height. For example, at location #12, the resulting return significant  wave heights for the return periods of 50-year, 100-year, 150-year and 200-year are 9.59 m, 9.86 m, 9.99 m and  10.06 m, respectively. However, under the criterion of Liang et al. (2019), the corresponding return significant  wave heights are 9.69 m, 9.89 m, 9.96 m and 10.05 m, respectively. Benefitting from a diagnostic process of  Liang et al. (2019), in our study we obtain return significant wave heights that are more stable than those of Shao et al. (2018a). Note that in table 9 of Shao et al. (2019) and tables 1 and 2 in this study, the return significant wave heights for the return periods of 50-year, 100-year, 150-year and 200-year are very similar at the same 22 study locations. Both groups of return significant wave heights are reasonable in the SCS. However, the threshold selection criterion in this study is suitable only in a tropical cyclone-dominated area.*

Two questions of mine about this paragraph:
a)  if yours and Shao et al. (2019) results are similar, what is the relevance of your method? Please explain
b)  Which feature of your method makes it suitable only in a tropical cyclone-dominated area? Please explain

- Second paragraph

I suggest the following  rephrasing of the first two sentences:
*The analysis of the initial database and the characteristics of the tropical cyclones determine a bimodal shape of the maximal SWH distribution. The threshold to be used in the POT method can be identified, without a subjective definition, as the value separating the two lobes of the maximum SWH distribution.*
The remaining three sentences (lines here below)  are not clear to me. A rephrasing is needed. To which part of the articles d they refer?
*A fixed distance is used to identify the initial database at the study site. This  fixed distance allows some small samples (the corresponding track is far, or the intensity is weak) to be extracted.  Thus, other analyses are needed to identify the extreme sample from the sample.*

- My understanding is the first three lines of this second paragraph describe your main conclusions. However, you may consider adding a new paragraph with short and clear description of 1) your main methodological conclusion and 2) the thresholds selection criterion

- Final Paragraph

*Consequently, the results of this study present a concept linking the assessment of extreme significant wave heights with the characteristics of tropical cyclones in a tropical cyclone-dominated area. The sample at the targeted location is affected by the track and intensity of the tropical cyclone. Future studies are suggested to  promote the assessment of extreme significant wave heights in a tropical cyclone. For example, the threshold may  be determined directly through a combination of track threshold and intensity threshold.*
Comment: This paragraph is rather obscure. The final sentence (which conclude the all article) suggests that the criterion proposed in this study will be likely revised in future and weakens the relevance of your work

- The abstracts should include a clear description of the proposed criterion for the threshold.

**Reviewer 3**

The manuscript improved a lot, and now the content of the study, and what the authors did, is much clearer. There are still issues with the English, and I would again suggest the manuscript to be proof read by a mother language. Below a list of minor comments.

pag 1, line 15: substitute excesses with exceedances

pag 3, line 6-7: this method makes the mosto of the samples ...., this sentence is unclear

pag 4, line 1: substitute everywhere maximal with maximum

pag 4, line 3: substutute "some unselected maximale swh" with "some unselected peaks of swh"

pag 4, line 7: substitute everywhere minimal with minimum

pag 4, line 18: substitute "(the acronym is ATSME)" with "(ATSME)"

pag 5, line 8: maximal -> maximum

pag 6, line 3: maximal -> maximum

pag 6, line 15: substitute excess with exceedance

pag 6, line 13: "return significant wave height for the i-year" is very unclear, substitute with "i-year return level of significant wave height", or "i-year return significant wave height".

pag 7, line 6: the first sentence in not entirely clear. Are u1..um the tested thresholds?

pag 7, line 7: substitute "return significant wave height for the i-year" with "i-years return level of significant wave height".

pag 8, line 12: I would suggest to reformulated paragraph 2.2, starting with and explanation of what ATSME is in plain language, and then write in detail the algorithmic processes.

pag 9, line 9: Is it the ERA-INTERIM reanalysis? Then write it and cite the references.

pag 9, line 9: After European Centre for ... add (ECMWF)

pag 9, line 16: the number is 247 to 403, from what do these different numbers come?

pag 11, line 16: the lower bound of the range is constant. Is it meaningful to select a threshold valid for all the return periods?

pag 12, 13, 14: maximal -> maximum

section 5: I understand that the bimodal distribution of Hs is a consequence of the sampling: the values beyond the separation will be, generally, the TC within 300km from the point, the ones below the TC beyond 300km selected for another point (correct?) So, I guess, a correct threshold should be higher than the separation between the 2 modes.

Wouldn't it be simpler to select for a given location just the TC within 300km from that point? Would this remove the bimodal shape and/or have impact on the performance of ATSME?

section 5: In my opinion there is still room for improvement in this section. It should be stated more explicitly, that the bimodal shape of the distribution is a result of the sampling technique, that a correct threshold should be beyond the separation, and that (if) the upper boundary of the ATSME range satisfies this requirement.

pag 18: I would include location 1 in table 2.

pag 18: in table 2 I would also indicate the value of the separation

**Reviewer 1**

The manuscript is significantly improved from the first submission and the authors should be commended for the level of detail at which they have addressed the reviewers' comments. The direction and flow of the paper is better, as is some of the technical explanation.

I still believe some of the paragraphs in the introduction can be broken down into smaller paragraphs for readability.

It would help if the headings for sections 2.1 and 2.2 were written in full rather than using acronyms.

Have the authors read any of the work by Young et al. regarding the relationships between tropical cyclones and wave fields? There is some very relevant information in his work that could be referenced in your paper. For example:

Young (2017). A review of parametric descriptions of tropical cyclone wind-wave generation. Atmosphere. 8(194).

Young and Vinoth (2013). An extended fetch model for the spatial distribution of tropical cyclone wind waves as observed by altimeter. Ocean Engineering, 70, 14-24.

My recommendation is for acceptance with minor revision.

---

## Author Response (AR2)

**Replies to comments by editor and reviewers**

**Extreme significant wave height of tropical cyclone waves in the South China Sea**

Zhuxiao Shao[1], Bingchen Liang[1, 2], Huajun Li[1, 2], Ping Li[3], Dongyoung Lee[1, 4]

[1] College of Engineering, Ocean University of China, 238 Songling Road, Qingdao 266100, China

[2] Shandong Province Key Laboratory of Ocean Engineering, Ocean University of China, 238 Songling Road, Qingdao 266100, China

[3] China Classification Society, Beijing 100007, China

[4] Korea Institute of Ocean Science & Technology, Ansan, Korea

Thank you for your letter and for your evaluation of our manuscript entitled "Extreme significant wave height of tropical cyclone waves in the South China Sea" (Ref: nhess-2018-349). Thanks for the recommendation of reviewer #2 and for the comments of reviewers #1 and #3. These comments are all valuable and very helpful for improving our paper. We have addressed all comments carefully and have made corrections corresponding to comments point by point. We hope that the improved manuscript will meet your approval.

The main corrections in the manuscript and responses to comments are shown as follows:

**Editor**

**(1) Page 18, line 4 replace "Conclusions and discussions" with "Discussion and conclusions"**

**Response:** As suggested, we have replaced "Conclusions and discussions" with "Discussion and conclusions".

**(2) I still consider the conclusions of your study not clearly written in the final section (see below).**

**Response:** As suggested, conclusions have been clearly rewritten in the final section.

**(3) First paragraph of the "Conclusions". Two questions of mine about this paragraph: a) if yours and Shao et al. (2019) results are similar, what is the relevance of your method? Please explain; b) Which feature of your method makes it suitable only in a tropical cyclone-dominated area? Please explain.**

**Response:** As suggested, the first paragraph has been rephrased. In this paragraph, we analyse the threshold selection criteria of Shao et al. (2018a) and Liang et al. (2019). Both of them select the suitable threshold within the stable threshold range. Benefiting from the stable characteristic of return significant wave heights, their threshold selection criteria can be used to assess the extreme significant wave height. Analysing the difference in these criteria: the first criterion is relatively simple; and the second criterion is relatively stable, due to a diagnostic process of return significant wave heights. When the variation of few return significant wave heights is relatively large in the stable threshold range, the return significant wave heights of Liang et al. (2019) are more stable than those of Shao et al. (2018a), especially for a short return period.

Liang et al. (2019) define the largest threshold of the stable threshold range as the suitable threshold for different return periods. In this study, we select the separation within the stable threshold range as a suitable threshold, depending on the characteristic of the

tropical cyclone wave. The suitable threshold is determined in the stable threshold range without a subjective definition. Considering that the sample distribution reflects the characteristic of the tropical cyclone wave, the threshold selection criterion is suitable in a tropical cyclone-dominated area.

- See the manuscript P. 19, lines 2-11: "In general, Shao et al. (2018a) and Liang et al. (2019) select the suitable threshold within the stable threshold range. Benefiting from the stable characteristic of return significant wave heights, their threshold selection criteria can be used to assess the extreme significant wave height. The first criterion is relatively simple; and the second criterion is relatively stable, due to a diagnostic process of return significant wave heights. For example, at location #12, Shao et al. (2018a) extrapolate the return significant wave heights for the return periods of 50-year, 100-year, 150-year and 200-year, which are 9.59 m, 9.86 m, 9.99 m and 10.06 m, respectively. However, under the criterion of Liang et al. (2019), the corresponding return significant wave heights are 9.69 m, 9.89 m, 9.96 m and 10.05 m, respectively. When the variation of few return significant wave heights is relatively large in the stable threshold range, the return significant wave heights of Liang et al. (2019) are more stable than those of Shao et al. (2018a), especially for a short return period."

**(4) Second paragraph of the "Conclusions". My understanding is the first three lines of this second paragraph describe your main conclusions. However, you may consider adding a new paragraph with short and clear description of 1) your main methodological conclusion and 2) the thresholds selection criterion**

**Response:** As suggested, the second paragraph has been rephrased to clearly show our thoughts. We add a short and clear description of our main methodological conclusion and threshold selection criterion.

- See the manuscript P. 19 and 20, lines 12-21 and 1-5: "To determine the suitable threshold within the stable threshold range without a subjective definition, the thresholds within the stable threshold range are further analysed, associating with the characteristic of the tropical cyclone wave. When studying the tropical cyclone wave, a fixed distance is used to identify the initial database at the study site. This fixed distance allows some small samples (the corresponding track is far, or the intensity is weak) to be extracted; however, no large samples (the corresponding track is close and intensity is strong) are neglected. Associated with these influences (i.e., track and intensity influences) of the tropical cyclones, the sample distribution has a natural separation distinguishing the high sample (a strong influence of the tropical cyclone) from the low sample (a weak influence of the tropical cyclone). Linking this distribution with the stable threshold range, the separation is within the stable threshold range. Thus, this separation can be used to identify the extreme sample (i.e., high sample in the distribution). Note that in Table 9 of Shao et al. (2019) and Tables 1 and 2 in this study, the return significant wave heights for the return periods of 50-year, 100-year, 150-year and 200-year are similar at the same 22 study locations. However, the threshold selection criterion in this study is relatively simple and objective, and this criterion can reflect the characteristic of the tropical cyclone wave. In addition, under this criterion, the asymptotic tail approximation and estimation uncertainty show that the fits are good and the uncertainties of the return significant wave heights are acceptable."

**(5) Final paragraph of the "Conclusions". This paragraph is rather obscure. The final sentence (which conclude the all article) suggests that the criterion proposed in this study will be likely revised in future and weakens the relevance of your work.**

**Response:** As suggested, the final paragraph is rewritten to clearly show our thoughts. The final sentence is rephrased to conclude the all article.

- See the manuscript P. 20, lines 6-10: "Considering that the sample distribution reflects the characteristic of the tropical cyclone wave, the threshold selection criterion is suitable in a tropical cyclone-dominated area. In this area, the initial database and characteristics of the tropical cyclones determine a bimodal shape of this distribution, which has a separation within the stable threshold range. Because the separation is objectively determined by the track and intensity of the tropical cyclone, this value can be identified as a suitable threshold in the POT method."

**(6) The abstracts should include a clear description of the proposed criterion for the threshold.**

**Response:** As suggested, a clear description of the proposed criterion for the threshold is included in the abstract.

- See the manuscript P. 1 and 2, lines 17-20 and 1-3: "To determine a suitable threshold, the sensitivity of return significant wave heights and the characteristics of tropical cyclone waves are studied. The sample distribution presents a separation that distinguishes the high sample from the low sample, and this separation is within the stable threshold range. Because return significant wave heights are stable in the stable threshold range and the separation is objectively determined by the track and intensity of the tropical cyclone, the separation is selected as a suitable threshold for extracting

the extreme sample in the tropical cyclone wave. The asymptotic tail approximation

and estimation uncertainty show that the selection is reasonable."

**Reviewer #3**

**The manuscript improved a lot, and now the content of the study, and what the authors did, is much clearer. There are still issues with the English, and I would again suggest the manuscript to be proof read by a mother language. Below a list of minor comments.**

**Response:** Thank you for your evaluation of our manuscript. As suggested, proofreading and language editing have been completed by American Journal Experts. In addition, we have addressed all comments carefully.

**(1) pag 1, line 15: substitute excesses with exceedances**

**Response:** As suggested, we have replaced "excesses" with "exceedances".

**(2) pag 3, line 6-7: this method makes the mosto of the samples ...., this sentence is unclear**

**Response:** As suggested, we have rephrased this sentence to show our thoughts clearly.

- See the manuscript P. 3, lines 10-12: "This method (i.e., the POT/GPD method) makes the most of the samples and extends the return period when the threshold is suitable (Alves and Young, 2003; You, 2011; Vanem, 2015a; Samayam et al., 2017; Shao et al., 2017), due to this method extracts all high samples."

**(3) pag 4, line 1: substitute everywhere maximal with maximum**

**Response:** As suggested, we have replaced "maximal" with "maximum".

**(4) pag 4, line 3: substutute "some unselected maximale swh" with "some unselected peaks of swh"**

**Response:** As suggested, we have replaced "some unselected maximal swh" with "some unselected peaks of swh".

**(5) pag 4, line 3: substitute everywhere minimal with minimum**

**Response:** As suggested, we have replaced "minimal" with "minimum".

**(6) pag 4, line 18: substitute "(the acronym is ATSME)" with "(ATSME)"**

**Response:** As suggested, we have replaced "(the acronym is ATSME)" with "(ATSME)".

**(7) pag 5, line 8: maximal -> maximum**

**Response:** As suggested, we have replaced "maximal" with "maximum".

**(8) pag 6, line 3: maximal -> maximum**

**Response:** As suggested, we have replaced "maximal" with "maximum".

**(9) pag 6, line 15: substitute excess with exceedance**

**Response:** As suggested, we have replaced "excess" with "exceedance".

**(10) pag 6, line 13: "return significant wave height for the i-year" is very unclear, substitute with "i-year return level of significant wave height", or "i-year return significant wave height".**

**Response:** As suggested, we have replaced "return significant wave height for the i-year" with "i-year return significant wave height".

**(11) pag 7, line 6: the first sentence in not entirely clear. Are u1..um the tested thresholds?**

**Response:** As suggested, this sentence is rephrased. The terms $u_1, \ldots, u_m$ are candidate thresholds (i.e., tested thresholds).

● See the manuscript P. 8, line 1: "The terms $u_1, \ldots, u_m$ are candidate thresholds."

**(12) pag 7, line 7: substitute "return significant wave height for the i-year" with "i-years return level of significant wave height".**

**Response:** As suggested, we have replaced "return significant wave height for the i-year" with "i-year return significant wave height".

**(13) pag 8, line 12: I would suggest to reformulated paragraph 2.2, starting with and explanation of what ATSME is in plain language, and then write in detail the algorithmic processes.**

**Response:** As suggested, the sequence of paragraphs in subsection 2.2 is adjusted. The explanation of the ATSME is in the first paragraph and the algorithmic processes are in the second and third paragraphs.

**(14) pag 9, line 9: Is it the ERA-INTERIM reanalysis? Then write it and cite the references.**

**Response:** The employed ECMWF is ERA-40 (Uppala et al., 2005) and ERA-Interim (Dee et al., 2011). We have written them and cited the references.

**(15) pag 9, line 9: After European Centre for ... add (ECMWF)**

**Response:** As suggested, we have added "(ECMWF)".

**(16) pag 9, line 16: the number is 247 to 403, from what do these different numbers come?**

**Response:** We have explained the number in the paper.

- See the manuscript P. 9, lines 14-18: "When the distance between the centre of the tropical cyclone and the study site is within 300 km, this tropical cyclone is recorded, and hourly significant wave heights simulated during this tropical cyclone are adopted as the initial database at the study site. At the 22 study sites, the number of recorded tropical cyclones is 247 to 403, and the annual mean number of recorded tropical cyclones is 6.175 to 10.075."

**(17) pag 11, line 16: the lower bound of the range is constant. Is it meaningful to select a threshold valid for all the return periods?**

**Response:** When the lowest threshold of the stable threshold range is selected as the suitable threshold, a threshold for different return periods can be uniquely determined, and

more samples are used to extrapolate return significant wave heights with weaker estimation uncertainties. When the median threshold of the stable threshold range is selected as the suitable threshold, the return significant wave height can be more robust. When the highest threshold of the stable threshold range is selected as the suitable threshold, the reliability of the return significant wave height can be greater. To guarantee the security of the design wave height and show the relationship between the stable threshold range and return period, Liang et al. (2019) selected the highest threshold of the stable threshold range as the suitable threshold for different return periods. In this study, a separation within the stable threshold range is selected as a suitable threshold, which is objectively determined by the track and intensity of the tropical cyclone.

**(18) pag 12, 13, 14: maximal -> maximum**

**Response:** As suggested, we have replaced "maximal" with "maximum".

**(19) section 5: I understand that the bimodal distribution of Hs is a consequence of the sampling: the values beyond the separation will be, generally, the TC within 300km from the point, the ones below the TC beyond 300km selected for another point (correct?) So, I guess, a correct threshold should be higher than the separation between the 2 modes.**

**Response:** In the paper, we have explained that the initial database and characteristics of the tropical cyclones determine a bimodal shape of the sample distribution. A fixed distance is used to identify the initial database at the study site. This fixed distance allows some small samples (the corresponding track is far, or the intensity is weak) to be extracted; however, no large samples (the corresponding track is close and intensity is strong) are neglected. Associated with these influences (i.e., track and intensity influences) of the tropical cyclones, the sample distribution has a natural separation distinguishing the high sample (a strong influence of the tropical cyclone) from the low sample (a weak influence of the tropical cyclone). Linking this distribution with the stable threshold range, the

separation is within the stable threshold range. Thus, this separation, rather than a higher value, is used to identify the extreme sample (i.e., high sample in the distribution).

**(20) Wouldn't it be simpler to select for a given location just the TC within 300km from that point? Would this remove the bimodal shape and/or have impact on the performance of ATSME?**

**Response:** A fixed distance is used to identify the initial database at the study site, rather than the extreme sample at the study site. The track and intensity of the tropical cyclone influence the wave; thus, both of them influence the extreme sample. For example, although the tracks of tropical cyclones Trami in 2001 and Wutip in 2007 are close to location #1, the intensities of these tropical cyclones are weak when these tropical cyclones influence the waves at location #1 (shown in Fig. 4). The samples during these tropical cyclones cannot be used to analyse the extreme wave.

The fixed distance allows some small samples (the corresponding track is far, or the intensity is weak) to be extracted; however, no large samples (the corresponding track is close and intensity is strong) are neglected. If this distance is too small, the numbers of high sample and low sample will decrease together; however, the bimodal shape still exists due to the case shown in Fig. 4, and the ATSME is feasible.

**(21) section 5: In my opinion there is still room for improvement in this section. It should be stated more explicitly, that the bimodal shape of the distribution is a result of the sampling technique, that a correct threshold should be beyond the separation, and that (if) the upper boundary of the ATSME range satisfies this requirement.**

**Response:** In the paper, we have explained that the initial database and characteristics of the tropical cyclones determine a bimodal shape of the sample distribution. A fixed distance is used to identify the initial database at the study site. This fixed distance allows some small samples (the corresponding track is far, or the intensity is weak) to be extracted; however, no large samples (the corresponding track is close and intensity is strong) are

neglected. Associated with these influences (i.e., track and intensity influences) of the tropical cyclones, the sample distribution has a natural separation distinguishing the high sample (a strong influence of the tropical cyclone) from the low sample (a weak influence of the tropical cyclone). Linking this distribution with the stable threshold range, the separation is within the stable threshold range. Thus, this separation, rather than a higher value, is used to identify the extreme sample (i.e., high sample in the distribution).

**(22) pag 18: I would include location 1 in table 2.**

**Response:** As suggested, statistics at location #1 are included in Table 2.

**(23) pag 18: in table 2 I would also indicate the value of the separation**

**Response:** As suggested, separation is indicated.

**Reviewer #1**

**The manuscript is significantly improved from the first submission and the authors should be commended for the level of detail at which they have addressed the reviewers' comments. The direction and flow of the paper is better, as is some of the technical explanation.**

**Response:** Thank you for your evaluation of our manuscript.

**(1) I still believe some of the paragraphs in the introduction can be broken down into smaller paragraphs for readability.**

**Response:** As suggested, the fourth paragraph in the introduction is broken down into two paragraphs for readability.

**(2) It would help if the headings for sections 2.1 and 2.2 were written in full rather than using acronyms.**

**Response:** As suggested, headings for subsections 2.1 and 2.2 have been rewritten.

**(3) Have the authors read any of the work by Young et al. regarding the relationships between tropical cyclones and wave fields? There is some very relevant information in his work that could be referenced in your paper. For example: Young (2017). A review of parametric descriptions of tropical cyclone wind-wave generation. Atmosphere. 8(194). Young and Vinoth (2013). An extended fetch model for the spatial distribution of tropical cyclone wind waves as observed by altimeter. Ocean Engineering, 70, 14-24.**

**Response:** We have read some papers of Young. We have cited some relevant papers (Alves and Young, 2003; Young et al., 2012; Young and Vinoth, 2013; Young, 2017; Ribal and Young, 2019) in our manuscript.

[revised manuscript text omitted]

---

## Author Response (AR3)

**Replies to comments by editor**

**Extreme significant wave height of tropical cyclone waves in the South China Sea**

Zhuxiao Shao[1], Bingchen Liang[1, 2], Huajun Li[1, 2], Ping Li[3], Dongyoung Lee[1, 4]

[1] College of Engineering, Ocean University of China, 238 Songling Road, Qingdao 266100, China

[2] Shandong Province Key Laboratory of Ocean Engineering, Ocean University of China, 238 Songling Road, Qingdao 266100, China

[3] China Classification Society, Beijing 100007, China

[4] Korea Institute of Ocean Science & Technology, Ansan, Korea

Thank you for your letter and for your evaluation of our manuscript entitled "Extreme significant wave height of tropical cyclone waves in the South China Sea" (Ref: nhess-2018-349). Thank you for your comments, which are all valuable and very helpful for improving our paper. We have addressed all comments carefully and have made corrections corresponding to comments point by point. We hope that the improved manuscript will meet your approval.

The main corrections in the manuscript and responses to comments are shown as follows:

**(1) - The manuscript lack a clear statement of the meaning of "stable threshold range" and "stable return significant wave heights". Please explain these concepts clearly in the introduction, before using them.**

> **Response:** As suggested, we have clearly stated the meanings of "stable threshold range" and "stable return significant wave height" in the introduction. In the sensitivity analysis of the return significant wave height to the threshold, the GPD model is fitted over candidate thresholds, and the suitable threshold is determined by studying the variation of return significant wave heights. When the return significant wave height is insensitive to

the threshold (i.e., the variation of return significant wave heights is generally small), the corresponding return significant wave height is defined as the stable return significant wave height, and the corresponding range of thresholds is defined as the stable threshold range.

- See the manuscript P. 4, lines 20-23: "When the return significant wave height is insensitive to the threshold (i.e., the variation of return significant wave heights is generally small), the corresponding return significant wave height is defined as the stable return significant wave height, and the corresponding range of thresholds is defined as the stable threshold range."

**(2) - Abstract: Line 18-22: rephrase the sentences "….. and this separation is within the stable threshold range. Because return significant wave heights are stable in the stable threshold range and the separation is objectively determined by the track and intensity of the tropical cyclone, the separation is selected as a suitable threshold for extracting the extreme sample in the tropical cyclone wave." In fact, while reading the abstract,it is not clear what "stable" specifically means**

**Response:** As suggested, we have associated the meaning of "stable" to the variation characteristic of return significant wave heights. If the variation of return significant wave heights is generally small, these return values are stable. As suggested in the first comment, clear statements about "stable threshold range" are presented in the introduction.

- See the manuscript P. 1, lines 20-22: "Because the variation of return significant wave heights in this range is generally small and the separation is objectively determined by the track and intensity of the tropical cyclone, the separation is selected as a suitable threshold for extracting the extreme sample in the tropical cyclone wave."

**(3) - Page 4 line 20-21: analogously, rephrasing is needed.**

**Response:** As suggested, we have rephrased the corresponding sentence.

- See the manuscript P. 4, lines 20-23: "When the return significant wave height is insensitive to the threshold (i.e., the variation of return significant wave heights is generally small), the corresponding return significant wave height is defined as the stable return significant wave height, and the corresponding range of thresholds is defined as the stable threshold range."

**(4) - Page 19 line 3.4 .It is not clear to what first and second criterion refer. I suggest (if my understanding is correct) to replace "the first criterion is relatively simple; and the second criterion is relatively stable" with ""the criterion of Shao et al. (2018a) is relatively simple; and the criterion of Liang et al. (2019) is relatively stable"**

[revised manuscript text omitted]